# Place-based science from Okinawa: 18th-century climate and geology recorded in Ryukyuan classical music

Justin T. Higa[1,2], June Y. Uyeunten[2], Kenton A. Odo[2]

[1]Department of Earth Sciences, University of Hawaiʻi at Mānoa, Honolulu, Hawaiʻi, USA

[2]Ryukyu Koten Afuso Ryu Ongaku Kenkyu Choichi Kai USA, Hawaiʻi Chapter, Honolulu, Hawaiʻi, USA

*Correspondence to*: Justin T. Higa (higa.justint@gmail.com)

**Abstract.** Indigenous knowledge can record scientific observations of specific "places" that may be difficult to preserve in the geologic record. Such place in "place-based" science highlights issues local to a learner for engaging audiences with the scientific problems relevant to their communities. Here, we focus on a popular seafaring repertoire of Indigenous Ryukyuan classical music, called Nubui Kuduchi and Kudai Kuduchi, to examine place-based observations of 18th-century climate and geology in the Ryukyu Islands (21st-century Okinawa Prefecture, Japan). By comparing the environmental conditions recorded in these songs with those of 20th–21st-century studies, we find that surface winds, ocean currents, typhoons, and volcanism from lyrics parallel their respective observations in the scientific record. This novel perspective of art and science highlights the relevance of Ryukyuan classical music in teaching contemporary issues such as climate change and natural hazards. Thus, Ryukyuan Indigenous knowledge can play an innovative role in science engagement for 21st-century Okinawans in Okinawa Prefecture and their diasporic kinsfolk worldwide.

## 1 Introduction

Indigenous knowledge can preserve geologic histories that are difficult to infer from the geologic record, thus guiding modern scientific inquiry. However, it was not until the late 20th century that researchers began incorporating such untapped data in environmental science literature (e.g., climate science; Petzold et al., 2020). Indigenous knowledge of plant, animal, and weather cycles have since been examined for signals of climate change and fortifications against climate hazards (e.g., Harmon et al., 2021; Hinzman et al., 2005; Hiwasaki et al., 2015; Turner and Reid, 2022). Workers also utilized Indigenous oral histories (e.g., Cascadia earthquakes in Ludwin et al., 2005), written records (e.g., Hawaiian hurricanes in Businger et al., 2018), and other sources of gray knowledge (e.g., various hazards of Aotearoa New Zealand in Bailey-Winiata et al., 2024) to improve modern-day disaster preparedness. Notably, scientists and cultural practitioners have focused on Indigenous visual and performing arts to examine historical volcanic activity (Swanson, 2008), earthquakes (Hough, 2007; Ludwin et al., 2005), and ecological resources (Gibson and Puniwai, 2006; Turner and Reid, 2022). Expanding the scientific analyses of artistic traditions to more Indigenous cultures can increase research objectivity and creativity by introducing new questions from these marginalized viewpoints (Bang et al., 2018; Intemann, 2009). Thus,

continued efforts in geoscience to incorporate Indigenous knowledge from art have great potential to 1) document more
historical records of climate and geological phenomena, 2) jumpstart new collaborations between cultural practitioners and
geoscientists, and 3) diversify scientific ways of knowing through the integration of Indigenous traditions.
In addition to research, highlighting Indigenous knowledge within a "place-based" framework allows educators and
students to better engage with the geosciences by tapping personal experiences of specific "places" from their local, natural
world (Semken et al., 2017). For example, workers in Hawai'i have tested how geoscience classes can increase engagement
with students by incorporating local indigenous stewardship, elder knowledge, Hawaiian place name meanings, and
Hawaiian language newspaper records into their curriculum (e.g., Chinn et al., 2014; Gibson and Puniwai, 2006). Efforts in
the Acoma Pueblo community of New Mexico, USA, have similarly integrated place-based concepts into education to teach
about local stratigraphy, hydrology, and natural resources (e.g., Reano and Hasara, 2024; Reano and Ridgway, 2015).
Furthermore, Palmer et al. (2009) focused on Indigenous art from the Southern Great Plains of North America to teach
undergraduate science modules across mineralogy, groundwater, and climate hazards. These educators have demonstrated
how rigorous programs that incorporate multiple ways of knowing can increase the interest, participation, and retention of
students from marginalized communities (Alexiades et al., 2021). Therefore, expanding place-based, artistic, and Indigenous
knowledge in geoscience engagement is likely key to addressing the needs of more Indigenous Peoples.
A promising candidate for science engagement with place-based Indigenous knowledge involves the Ryukyuan
Peoples from the Ryukyu Islands, comprising 21st-century Okinawa Prefecture, Japan. Previous works have connected
Ryukyuan Indigenous knowledge with broad geoscience topics such as groundwater resources (Takahashi, 2022), regional
fisheries (Toguchi et al., 2016), and coral reef geobiology (Toguchi and Nishime, 2013). However, to the best of our
knowledge, no works in English or Japanese have examined the connections between geoscience and one of the most
influential Indigenous art forms across Okinawa and the Okinawan diaspora, Ryukyuan classical music, also known as
*Ryūkyū koten ongaku* (琉球古典音楽; hereafter RKO). We compare this tradition with contemporary science across the
atmo-, hydro-, and geosphere to address the question: Does RKO record place-based environmental phenomena useful for
geoscience engagement? Here, we focus on a popular RKO repertoire, Nubui Kuduchi (上り口説) and Kudai Kuduchi (下
り口説), to trace the 18th-century experiences of Ryukyuan seafaring envoys between Okinawa and Kyushu islands
(Okinawa Prefectural Cultural Promotion Association, 2001; Toby, 1984). A review of contemporary geoscience literature
finds parallels between these songs and science in Ryukyuan surface winds, oceanic circulation, typhoon activity, and
volcanism. Such similarities highlight how Nubui Kuduchi and Kudai Kuduchi may record benchmarks of historical
environmental conditions and how Ryukyuan Indigenous knowledge can connect a prominent art form with complex
environmental research. Educators and cultural practitioners may use these insights in place-based geoscience engagement,
where demand for this work is likely high due to an active Ryukyuan arts scene across the world. Accordingly, we showcase
how popular Indigenous music can document scientific observations of climate and geology to engage Indigenous Peoples
with their contemporary environmental heritage.

**1.1 Ryukyu Kingdom, Okinawa Prefecture, and the Okinawan diaspora**

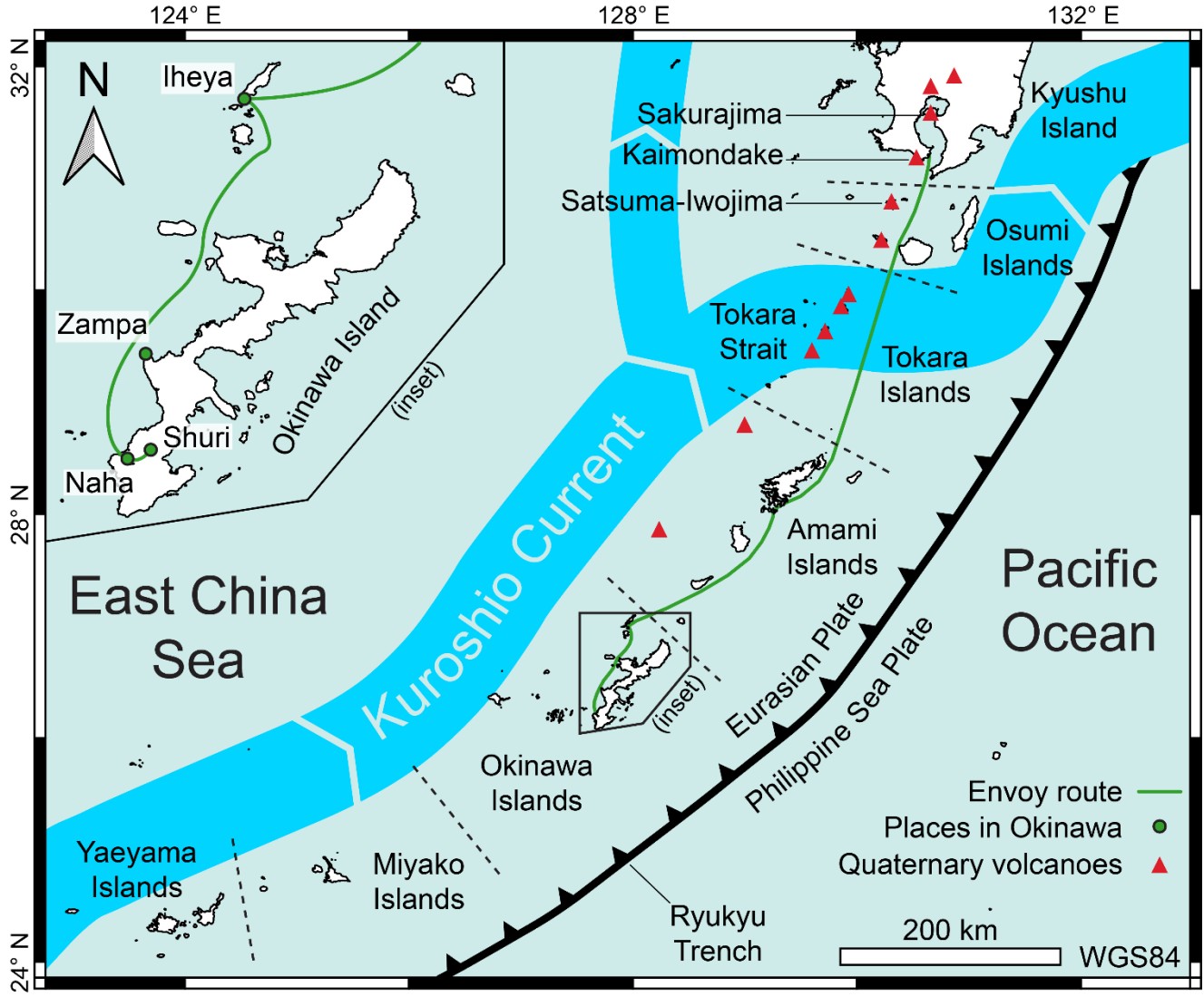

**Figure 1: The Ryukyu Islands and southern Kyushu. Map shows the Kuroshio Current (Gallagher et al., 2015), Ryukyu Trench (Kamata and Kodama, 1994), approximate route of Ryukyuan envoys (Okinawa Prefectural Cultural Promotion Association, 2001), and subaerial Quaternary volcanoes (Global Volcanism Program, 2024). (inset) Detailed map of Okinawa Island. Geography from U.S. Department of State, Office of the Geographer (2013).**

The Ryukyu Islands span a north-south transect between Kyushu and Taiwan in the western Pacific Ocean, encompassing the Osumi, Tokara, Amami, Okinawa, Miyako, and Yaeyama islands at its maximum geographical extent (Fig. 1). This island arc is the ancestral home of the Indigenous Ryukyuan People and the former Ryukyu Kingdom, established in the 15$^{th}$ century and centered on Okinawa Island (Sakiyama and Oshiro, 1995; Toby, 1984). The kingdom colonized south to the Yaeyama Islands and north to the Tokara Islands during the height of this dynastic period (Akamine, 2017). Contact

from foreign trade influenced Ryukyuan culture, including from China, Japan, Korea, Thailand, Malaysia, and Indonesia
(Sakiyama and Oshiro, 1995). However, in 1609 CE, the Ryukyu Kingdom was invaded and subsequently controlled by
Japanese forces in the Satsuma Domain of southern Kyushu and the Tokugawa Shogunate in Edo (pre-1868 CE name for
Tokyo; Akamine, 2017; Toby, 1984). Consequently, most Ryukyuan territory north of Okinawa Island was ceded to the
Satsuma Domain (Akamine, 2017). Historical records suggest ~20 Ryukyuan envoys traveled between Okinawa, Satsuma,
and Edo to pay tribute to the Shogunate from 1610 CE until 1872 CE (Okinawa Prefectural Cultural Promotion Association,
2001; Toby, 1984), followed by the annexation of the Ryukyu Kingdom by Japan as Okinawa Prefecture in 1879 CE
(Akamine, 2017). During this colonial period, the Japanese government employed assimilationist education policies with the
goal of eliminating Ryukyuan languages and cultures (Hammine, 2019; Kaneshiro, 2002). This policy lasted until 1945 with
the end of World War II and the start of occupation by the USA (Hammine, 2019). Since the reversion of Okinawa
Prefecture from the USA back to Japan in 1972, Ryukyuan culture experienced a resurgence across grassroots movements
(e.g., Inoue, 2004), language revitalization efforts (e.g., Heinrich, 2018; Zlazli, 2021), and statements from the Okinawan
prefectural government (e.g., Abe, 2023).
Despite this marginalization, the Ryukyu sphere of influence expanded out of East Asia during 19th–20th-century
emigration from Okinawa Prefecture, namely to Hawai'i, Brazil, and Peru that received roughly 20,000, 15,000, and 11,000
immigrants by 1938, respectively (Sellek, 2003). Estimates place cumulative Okinawan immigration between 150,000–
200,000 people before the end of World War II; immigrants to the Japanese mainland were mostly comprised of factory
workers, whereas those abroad worked on plantations (Roberson, 2010; Sellek, 2003). In Japan, Okinawans faced a similar
system of discrimination as in Okinawa, compounded by social isolation, low wages, and dangerous working conditions
(Roberson, 2010). Migrants overseas faced discrimination from two sources: poor plantation conditions by the elite-class
plantation owners and a previously established, Japanese immigrant community that held similar prejudices as in Japan
(Kaneshiro, 2002; Kodama, 1981; Ueunten, 1989). In Hawai'i, such racial tensions continued until roughly the end of World
War II, when second-generation Okinawan and Japanese Americans became dominant over the first-generation immigrants;
shared experiences in plantation labor unions, American military service, and communal education likely led to the gradual
relinquishment of former prejudices (Ueunten, 1989). Nonetheless, the initial separation of Okinawan and Japanese led to a
distinct, Okinawan, and diasporic identity. This identity is evident in the establishment and success of the Hawai'i United
Okinawa Association (HUOA), an amalgamation of ~50 affinity groups that support Okinawan culture and community in
Hawai'i (Kaneshiro, 2002; Kodama, 1981; Ueunten, 1989). An example of HUOA's success is the annual Okinawan
Festival, one of the largest cultural events in Hawai'i that attracts ~50,000 attendees in the 21st century (Taira, 2023).
Brazilian- (Mori, 2003) and Argentine-Okinawans (Alonso Ishihara, 2022), as well as Okinawan communities across the
USA (Okamura, 2022), founded similar associations. Moreover, HUOA and these other associations participate in the
Worldwide Uchinanchu Festival hosted by Okinawa Prefecture (Uchinanchu means "Okinawan People" in one Ryukyuan
language); overseas associations sent ~8,000 attendees in 2016 (Okamura, 2022). Such efforts are fueled by a "Born Again

Uchinanchu" movement that encapsulates third and later generations of the Okinawan diaspora looking to reconnect with their culture and heritage (Chinen, 2025). Accordingly, Okinawan identity remains visible and active worldwide.

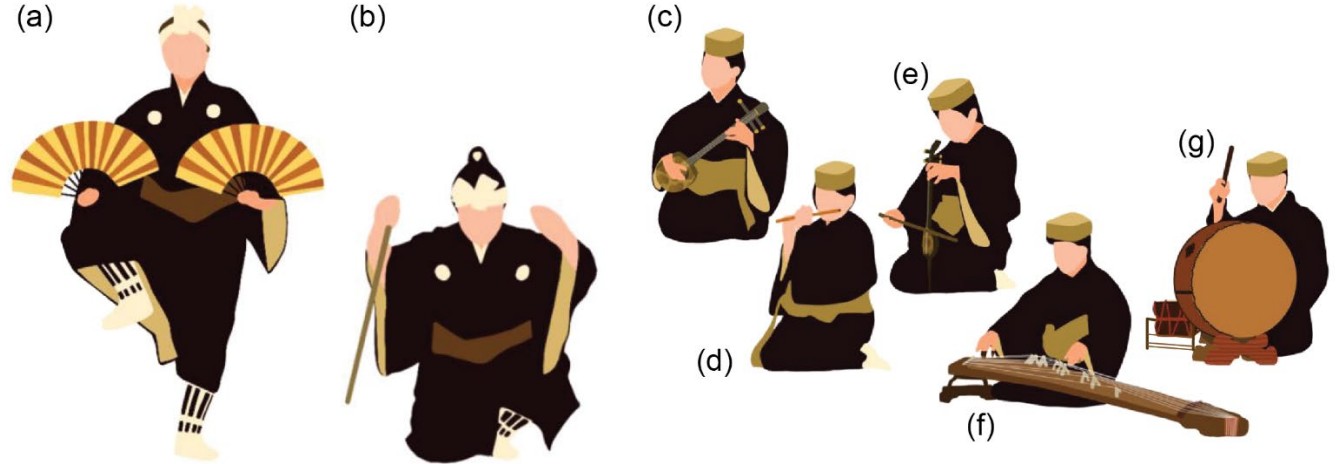

**Figure 2: Illustrations of (a) Nubui Kuduchi and (b) Kudai Kuduchi dancers with (c) *uta sanshin*, (d) *fwansō*, (e) *kūchō*, (f) *kutū*, and (g) *tēku* musicians. Actual performances may have more than one dancer or musician per instrument, particularly for *uta sanshin*. Illustration by B. Kuhasubpasin.**

### 1.2 *Ryūkyū koten ongaku* (RKO) and Indigenous knowledge

RKO is one of many visible cultural identifiers for Okinawans in Okinawa Prefecture and in the diaspora (Gillan, 2016; Kaneshiro, 2002; Teruya, 2014; Ueunten, 1989, 2020). According to Sakiyama and Oshiro (1995), RKO is an aristocratic genre that developed during the Ryukyu Kingdom's dynastic period for entertaining visiting emissaries, historically accompanied by male dancers from the noble class (Fig. 2a–b). RKO lyrics are originally from Ryukyuan poetry, which often focuses on metaphors of the natural world to convey human emotions and experiences. These performances are led by *uta sanshin* (唄三線), or a three-stringed lute with vocals (Fig. 2c). The *sanshin* lute itself was brought to Okinawa from China and became a symbol of Okinawan identity (Gillan, 2016). Nonetheless, the *uta* vocal component holds the melody of most RKO songs and is often said to be more central to RKO than *sanshin* (Ueunten, 2020). Other instruments that accompany *uta sanshin* include *fwansō* (笛; bamboo flute; Fig. 2d), *kūchō* (胡弓; fiddle; Fig. 2e), *kutū* (箏; zither; Fig. 2f), and *tēku* (太鼓; drums; Fig. 2g). Such RKO gained distinct Japanese influences after the Satsuma invasion and increased contact with the Satsuma Domain through Ryukyuan envoys (Okinawa Prefectural Cultural Promotion Association, 2001; Toby, 1984). RKO developed into commercial and popular theater when the demand for Ryukyuan court music collapsed after Japanese annexation (Gillan, 2016), which ended historical class and gender restrictions in these aristocratic arts. As such, 21[st]-century RKO performing arts schools have wide participation in Okinawa Prefecture and the Okinawan diaspora; these schools remain the main mode of RKO transmission to new learners (e.g., Gillan, 2016; Hanashiro, 2007; Kaneshiro, 2002; Ueunten, 1989). RKO is also visible to non-artists as stone monuments to transformative songs, called *kahi* (歌碑).

These monuments are often installed where songs have some lyrical or historical connection to a place, functioning as community centers, artistic venues, and memorials to collective Okinawan experiences (e.g., World War II) or as landmarks in popular *kahi* tours (Gillan, 2017). Furthermore, RKO gained national and prefectural support through designations of National Living Treasures by the Japanese government and the establishment of institutions such as the Okinawa Prefectural University of Arts and the National Theatre Okinawa (Gillan, 2016). Thus, RKO remains a vibrant marker of Ryukyuan culture across the Okinawan community.

Indigenous knowledge of climate and geology may be preserved in RKO, similar to that preserved in traditional ecological practices across the Ryukyu Islands (i.e., Takahashi, 2022; Toguchi et al., 2016; Toguchi and Nishime, 2013). Such artistic knowledge may be adapted into place-based science engagement that addresses issues specific to 1) Okinawans in Okinawa Prefecture and 2) diasporas across the world. First, subtropical Okinawa Prefecture will likely face challenges due to 21st-century anthropogenic climate change, including coastal flooding, typhoon intensification, and coral bleaching (IPCC, 2023). Likewise, the prefecture faces a stagnant college matriculation rate and low standardized test scores relative to the whole of Japan (Kakazu, 2012), which follows historical marginalization (e.g., Abe, 2023; Hammine, 2019; Inoue, 2004). Research in science education finds that such marginalization can preclude the next generation of marginalized people from entering environmental science studies and careers (Martin and Fisher-Ari, 2021; Padgett, 2001). Due to RKO's visibility in Okinawan culture, this art form may be key to geoscience engagement and retention efforts in Okinawa Prefecture (Semken et al., 2017). Second, RKO and other folk music genres serve as pillars of Okinawan identity for the Okinawan diaspora, separated from Okinawa by three or more generations in the 21st century and interested in ways to express their identity (Kaneshiro, 2002; Ueunten, 1989). These later generations may not understand Ryukyuan languages and RKO lyrics, which poses a barrier to accessing Ryukyuan Indigenous knowledge (Chinen, 2025). However, most RKO instructors in the diaspora have trained in or are from Okinawa Prefecture, increasing direct access to such knowledge from the homeland (Chinen, 2025; Kaneshiro, 2002; Miyashiro, 2018; Teruya, 2014; Ueunten, 1989). The activity in Okinawan associations like HUOA exemplifies a demand for more access to RKO- and place-based Indigenous knowledge. As such, our investigation of RKO can fulfill geoscience engagement goals in Okinawa Prefecture and the global diaspora by elevating place-based, Ryukyuan, Indigenous knowledge as a reputable way of knowing.

## 2    Methods

### 2.1 Song background and lyrical interpretation

We focus on Nubui Kuduchi and Kudai Kuduchi, which were composed during the Satsuma Domain's rule over the Ryukyu Kingdom. These songs are usually attributed to the RKO master Yakabi Chōki (屋嘉比朝寄; 1716–1775 CE; surname first following Japanese naming convention; Gillan, 2012; Kinjo, 1992). *Kuduchi* (口説) refers to a subgenre of RKO with a distinctly Japanese, rather than Ryukyuan, seven-five beat structure (Kinjo, 1992) and often tells a chronological

story (Seki, 2024). Then, *nubui* (上り) refers to "climbing up" to Satsuma and *kudai* (下り) to "climbing down" to Okinawa Island (Kinjo, 1992; Seki, 2024). Thus, these songs detail a Ryukyuan envoy's 18th-century journey between the Ryukyu Kingdom and Satsuma Domain during the Japanese colonial period (Fig. 1). Such performances were historically reserved for entertaining Satsuma Domain officials in the Ryukyu Kingdom, where performers dance with a folding fan in each hand or a traveler's cane for Nubui Kuduchi or Kudai Kuduchi, respectively (Fig. 2a–b; Sakiyama and Oshiro, 1995). The dance represents different aspects of the envoy and has a relatively masculine connotation related to its brisk tempo, karate influence, and the harrowing journey itself (Kinjo, 1992). Both songs remain popular in 21st-century RKO performances for entertainment and cultural preservation (Hanashiro, 2007).

Here, we create English synopses of Nubui Kuduchi and Kudai Kuduchi to scientifically interpret both songs. We utilize a version of these songs from the Afuso Ryū (安冨祖流) school of RKO (one of two major schools; Garfias, 1993; Gillan, 2012), alongside interpretations from Kinjo (1992), Sakiyama and Oshiro (1995), and Seki (2024). Following best practices in Younging (2018), the authors here include RKO Master Instructors June Y. Uyeunten and Kenton A. Odo (hereafter J.Y. Uyeunten and K.A. Odo, respectively) of the Ryukyu Koten Afuso Ryu Ongaku Kenkyu Choichi Kai USA (hereafter Choichi Kai USA), serving the Okinawan diaspora in Hawai'i, USA. Both authors provide access to oral and written information on Nubui Kuduchi and Kudai Kuduchi, including personal communications and interpretations from Clarence T. Nakasone (hereafter C.T. Nakasone; 1998) of the Hooge Ryu Hana Nuuzi no Kai Nakasone Dance Academy, also based in Hawai'i. In addition, the first author is an *uta sanshin* practitioner with Choichi Kai USA at the time of publication. We provide supplementary videos with song lyrics, translations, and interpretations from the above sources, with permission and production from J.Y. Uyeunten, K.A. Odo, and the aforementioned dance academy (Higa et al., 2024a, b). We caution that these supplements are solely to provide references for lyrical synopses; we do not claim intellectual property for the songs and lyrics and do not assert that these songs should enter the public domain or become *gnaritas nullius* ("no one's knowledge" in Latin; i.e., Younging, 2018). These precautions are to ensure that Indigenous knowledge is properly credited and utilized. In addition, we acknowledge that the other major RKO school, Nomura Ryū (野村流), may hold different versions of Nubui Kuduchi and Kudai Kuduchi. However, as all authors are members of an Afuso Ryū branch, we opt to use our school's version as a base, which is supplemented by insights from other textual and academic sources (i.e., Kinjo, 1992; Sakiyama and Oshiro, 1995; Seki, 2024). We note that Afuso Ryū and Nomura Ryū diverged from two students of the same Master Instructor in the 19th century and that most differences between the schools occur in singing or playing style; Afuso Ryū techniques are said to be closer to the original lineage, whereas those of Nomura Ryū were simplified and standardized to make the art more accessible (Garfias, 1993; Gillan, 2012). As the lyrics are mostly consistent across schools (except where we indicate notable variations), differences in interpretation due to version control are likely minimal.

## 2.2 Climate and volcanology literature review

We link observations of climate, geology, and the environment within Nubui Kuduchi and Kudai Kuduchi synopses to 20th- and 21st-century scientific studies by noting similarities and differences therein, similar to Swanson (2008). Swanson (2008) utilizes a Hawaiian oral history of the volcano deity Pele and her sister Hiʻiaka in combination with 1812 CE written records from European Christian missionaries to improve scientific interpretations of caldera formation at Kīlauea Volcano, Hawaiʻi. The version of the oral history examined in Swanson (2008) is believed to be the most unaffected by Western influences, but other versions are likely to be fundamentally similar, much like Nubui Kuduchi and Kudai Kuduchi are similar across RKO schools. Swanson (2008) presents a literature review on Kīlauea geochronology and stratigraphy to compare with the oral and written histories. It was found that radiometric dating, Hawaiian oral tradition, and written records agree that the extant Kīlauea Caldera likely formed between 1470–1500 CE; the caldera was previously thought to have formed in 1790 CE during an explosive eruption. A major discussion point of Swanson (2008) is that such a conclusion may have been accepted by the scientific community earlier if Indigenous knowledge had been seriously considered.

Here, we follow the methods of Swanson (2008) because we have a parallel aim to examine the correspondence between an Indigenous record and scientific literature. For our literature review, we cover 20th – 21st-century research across Ryukyuan climate and volcanology, two topics that we identify as likely recorded in Nubui Kuduchi and Kudai Kuduchi. We focus our review on the natural conditions of climate and volcanic systems that Ryukyuan envoys may have experienced during 18th-century travels. We also review potential impacts on 21st-century Okinawans by anthropogenic climate change or volcanic hazards. Next, we systematically extract lyrical observations from our synopses as either climate- or volcanology-related. We then subgroup these observations into specific phenomena or locations from the corresponding scientific literature to showcase similarities or differences between sources. Finally, we discuss novel scientific and cultural implications from such climate and volcanic links, which can be tailored to geoscience engagement in Okinawa Prefecture and abroad. We acknowledge that, unlike Swanson (2008), Nubui Kuduchi and Kudai Kuduchi do not point to a single geologic event, but rather to general climate or geologic conditions. Nonetheless, these generalized links can be used in lessons to relate RKO with environmental research. We therefore demonstrate the utility of Indigenous knowledge from RKO to increase geoscience engagement in Okinawan communities.

For the previous and following descriptions, we italicize Ryukyuan and Japanese common nouns and utilize diacritical marks for Ryukyuan and Japanese words where kana and kanji scripts are provided. Some kana may not reflect 21st-century Japanese pronunciation but represent common transliterations of the Ryukyuan languages. We use anglicized names and roman type for proper nouns or when kana and kanji are not provided.

## 3 Results

### 3.1 Lyrical synopses

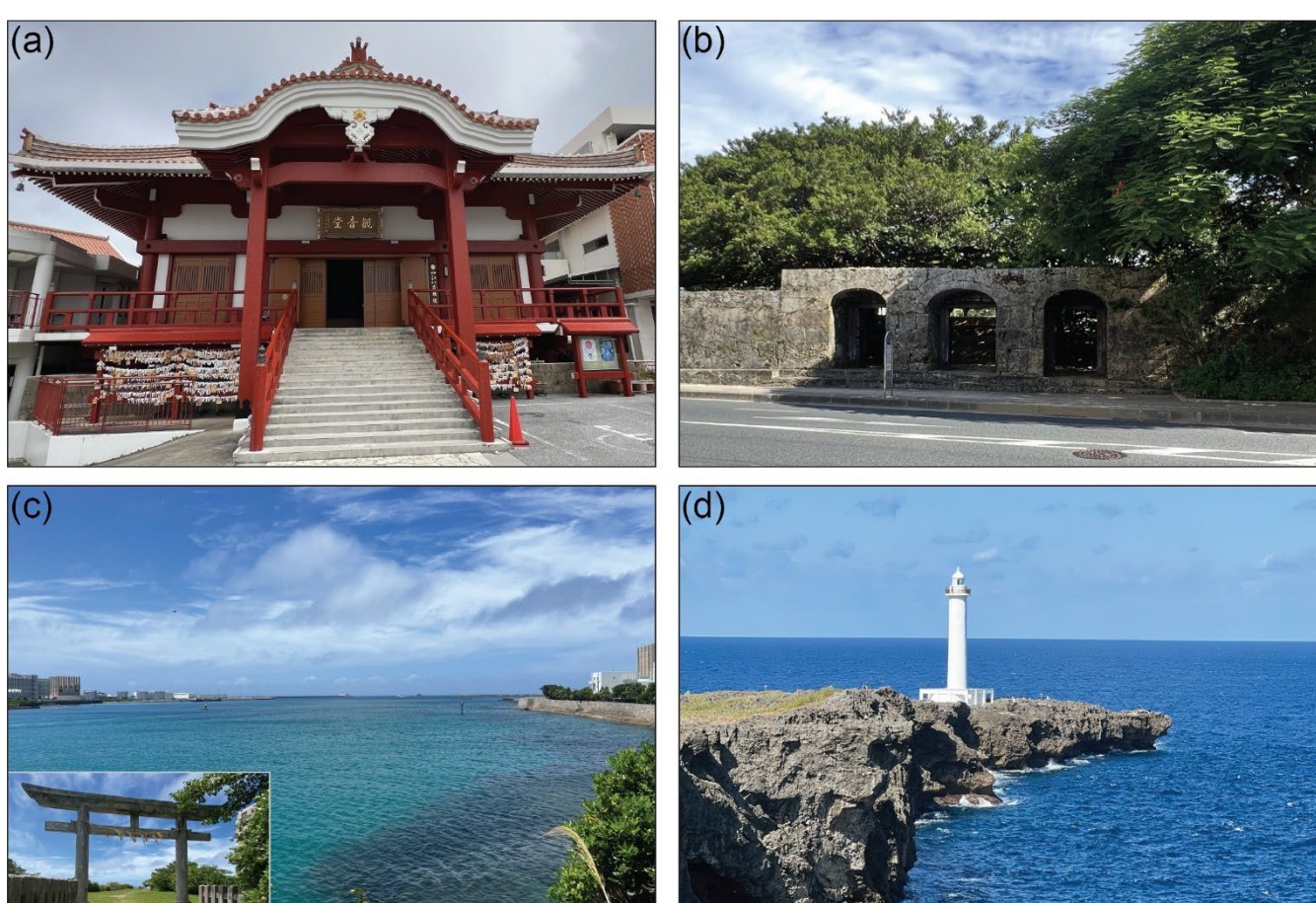

**Figure 3: Historical sites in Nubui Kuduchi and Kudai Kuduchi on Okinawa Island photographed from 2023–2024. (a) Shuri Kannondo Buddhist temple in Shuri from first verse of Nubui Kuduchi and (b) Sogenji Buddhist temple ruins in Naha from second verse of Nubui Kuduchi, where voyagers prayed for ocean safety. (c) View from Miegusuku Fortress in Naha from both songs, looking west out of Naha port where voyagers departed from and returned to Okinawa Island and (inset) atop fortress ruins. (d) Cape Zampa from both songs, looking west near where voyagers noted wind and ocean conditions. See locations in Fig. 1.**

Based on Kinjo (1992), Sakiyama and Oshiro (1995), and Seki (2024), discussions with J.Y. Uyeunten and K.A. Odo of Choichi Kai USA, and personal communications with C.T. Nakasone, Nubui Kuduchi details the perilous journey from Okinawa Island to Kyushu. K.A. Odo describes how the word *nubui* also references the difficulty and anxiety of leaving home as an upward climb. The first four verses describe the envoy's foot journey from Shuri, the Ryukyu capital at the time these songs were composed, to the main port on Okinawa Island in Naha (Fig. 1). Along the way, voyagers pray at and pass by various Ryukyuan shrines and Buddhist temples for safe travels, including Shuri Kannondo and Sogenji (Fig. 3a–b). Families of the envoy shed tears due to the dangers at sea facing their loved ones. The fifth and sixth verses detail the

start of the voyage when sails pick up winds from the south-southwest and the envoy travels out of Naha port, vowing to someday return to the Miegusuku Fortress (Fig. 3c) and Cape Zampa on the west coast of Okinawa Island (Fig. 3d). Here, south-southwest is referred to as the direction of the Horse and Sheep, collectively assigned this cardinal direction from the Chinese Zodiac (Seki, 2024). In the seventh verse, the envoy encounters rough seas near Iheya Island, ~100 km from the port but only ~40 km from the north point of Okinawa Island. Looking out over a "route of many islands" (i.e., Amami Islands; Okinawa Prefectural Cultural Promotion Association, 2001), the envoy surveys the upcoming seven islands (i.e., Tokara Islands in the Tokara Strait) that are often rough sailing and expresses hope for a peaceful transit. The last verse describes the final approach to the Satsuma Domain of southern Kyushu (Fig. 4a), wherein the envoy observes smoke from Iwodake volcano on the island of Satsuma-Iwojima (Fig. 4b). Of note, the version of this song from C.T. Nakasone describes the smoke as rising (*tachuru*; 立ちゆる), while the Afuso Ryū version notes the smoke is burning or glowing (*moyuru*; 燃ゆる). Finally, the envoy sails past Cape Sata on Kyushu, where Kaimondake (Fig. 4c) and Sakurajima (Fig. 4d) volcanoes come into view. The song ends with Sakurajima being hailed as mistakeable for the iconic Mount Fuji.

Based on Sakiyama and Oshiro (1995) and Seki (2024), discussions with J.Y. Uyeunten and K.A. Odo, and personal communications with C.T. Nakasone, verse one of Kudai Kuduchi begins when envoy members are called to return to Okinawa Island from the Satsuma Domain around the ninth to the tenth month of the lunar calendar (approximately September to October). Such travels occur after the envoy transits from Satsuma to Edo and back, which is not recorded in this repertoire. As *kudai* refers to the act of climbing down, K.A. Odo notes the subtext of an easier, downhill, and more hopeful return home. Consequently, joyous celebrations, more prayers for safety, and Satsuma bureaucratic processes are described in the first to fifth verses. In the sixth to eighth verses, the sails pick up north-northeasterly winds after passing Cape Sata, the rough seas of the Tokara Islands, the Amami Islands, and Iheya Island. North-northeast is referred to as the direction of the Rat and Ox zodiacs (Seki, 2024). Here, the envoy is accompanied by friendly vessels when the voyagers return to Cape Zampa, as promised (Fig. 3d). The final verse describes arriving home, with crowds of people welcoming the voyagers at Miegusuku Fortress (Fig. 3c).

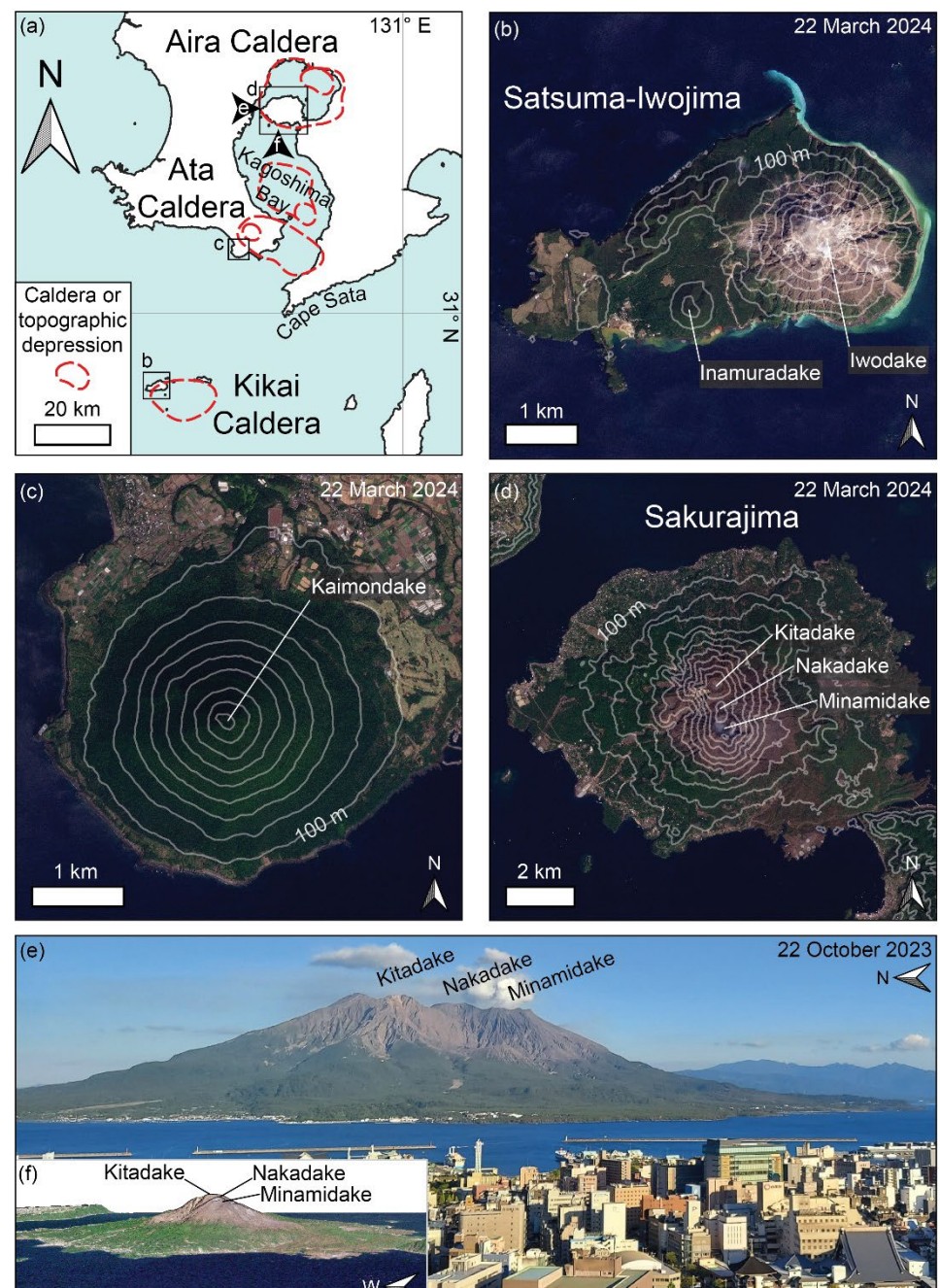

**Figure 4: (a)** Calderas and topographic depressions of southern Kyushu and the northern Ryukyu island arc (Maeno and Taniguchi, 2007; Nagaoka, 1988) with location of subplots as boxes or arrows indicating view direction. Geography from U.S. Department of State, Office of the Geographer (2013). PlanetScope 3 m resolution images of **(b)** Satsuma-Iwojima, **(c)** Kaimondake, and **(d)** Sakurajima (Image © 2025 Planet Labs; Planet Team, 2025) with 100 m elevation contours from 1 arc-second Shuttle Radar Topography Mission. **(e)** Sakurajima with Kagoshima City in foreground looking east versus **(f)** looking north with PlanetScope image draped over elevation.

## 3.2 Climate

### 3.2.1 Ryukyu climate review

The East Asian monsoon system drives atmospheric circulation and a seasonal transition of winds in the Ryukyu Islands, from northeasterlies during the Boreal winter to southwesterlies during the Boreal summer (Dobby, 1945; Flohn, 1957; Fu et al., 1983; Ueda et al., 1995). Then, the Kuroshio Current dominates oceanic circulation, bringing equatorial waters north into the East China Sea following the Ryukyu island arc; this warm current reenters the Pacific Ocean through the Tokara Strait and into the northern Pacific gyre (Fig. 1; Gallagher et al., 2015). Measurements of the Kuroshio Current along the Ryukyu Islands and through the Tokara Strait show high wave heights and ocean turbulence due to ocean-atmosphere coupling (Hwang, 2005) and interactions with bathymetric features such as volcanic seamounts along the Tokara Islands (Nagai et al., 2021; Tsutsumi et al., 2017). Previous works highlight how the East Asian monsoon and Kuroshio Current, thus the climate of the Ryukyu Islands, may be altered by anthropogenic climate change. For example, monsoonal winds (Kitoh, 2017) and Hadley cell circulation (Lu et al., 2007) may weaken over the western Pacific. Likewise, the Kuroshio Current is vulnerable to shifts in location and strength (e.g., Sakamoto et al., 2005; Wu et al., 2012; Zhang et al., 2020).

Typhoons are also common occurrences across the Ryukyu Islands (e.g., Ikema et al., 2010), constituting a critical feature of local climate and weather. Oceanic moisture (Ikema et al., 2010; Kitoh, 2017) and atmospheric forcing (Sun et al., 2015, 2017) help form then steer these storm systems from the tropical western Pacific toward subtropical East Asia (e.g., Wu and Wang, 2004; Yang et al., 2020). In turn, typhoons impact ocean conditions through increased wind speed and wave heights in the western Pacific region (Wu et al., 2014; Young et al., 2011). Moreover, the El Niño-Southern Oscillation (ENSO) can affect typhoon tracks. El Niño (i.e., periods of warming sea surface temperatures) is correlated with more typhoons recurring north, often toward the Ryukyu Islands; conversely, La Niña (i.e., periods of cooling sea surface temperatures) is characterized by more storms tracking west toward continental Asia (Ito et al., 2020; Sun et al., 2015, 2017; Wang and Chan, 2002; Wu and Wang, 2004; Yang et al., 2020). Variability in the western North Pacific Subtropical High is important for such tracking, as an eastward retraction of this persistent zone of high pressure allows typhoons to recurve north (Sun et al., 2015, 2017). These typhoons heading to higher latitudes help facilitate poleward energy transport (Wang and Chan, 2002). Therefore, while typhoons are weather phenomena, they both impact and are impacted by climate (e.g., ENSO). Typhoons are expected to become larger (Sun et al., 2015, 2017), rainier (Ikema et al., 2010; Kitoh, 2017), and more north-recurving (e.g., Yang et al., 2020) due to anthropogenic climate change.

## (a) Climate

**Lyrical observations**

Voyages from Okinawa to Kyushu during summer south-southwesterlies
• Nubui Kuduchi verse 5

Voyages from Kyushu to Okinawa in the ninth or tenth lunar month
• Kudai Kuduchi verse 1

Winter north-northeasterlies
• Kudai Kuduchi verse 6

Prayers for safe travels (Fig. 3a–b), vows to return to specific landmarks (Fig. 3c–d)
• Nubui Kuduchi verses 1, 2, 4, 6
• Kudai Kuduchi verses 2, 8, 9

Rough seas between Iheya and the Amami Islands
• Nubui Kuduchi verse 7
• Kudai Kuduchi verse 8

Famously dangerous Tokara Strait
• Nubui Kuduchi verse 7
• Kudai Kuduchi verse 6

### East Asian Monsoon

**Scientific literature**

Voyages from Okinawa to Kyushu occurred around summer (Okinawa Prefectural Cultural Promotion Association, 2001; Toby, 1984) when the Boreal monsoon is characterized by south-southwesterlies (Dobby, 1945; Flohn, 1957; Fu et al., 1983; Ueda et al., 1995).

Voyages from Kyushu to Okinawa occurred between autumn to spring when the Boreal monsoon is characterized by north-northeasterlies (Dobby, 1945; Fu et al., 1983).

### Ocean conditions

Western Pacific Ocean is a typhoon hotspot during summer to autumn months when envoys were active (e.g., Ito et al., 2020).

Envoy path has statistically high wave heights, partially due to typhoon activity (Wu et al., 2014; Young et al., 2011).

Kuroshio current interactions with atmosphere and bathymetry in the Tokara Strait may induce ocean turbulence (Fig. 1; Hwang, 2005; Nagai et al., 2021; Tsutsumi et al., 2017).

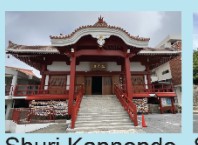 Shuri Kannondo (Fig. 3a)

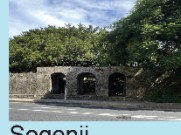 Sogenji (Fig. 3b)

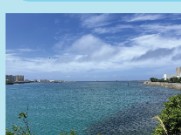 Miegusuku (Fig. 3c)

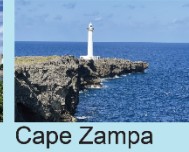 Cape Zampa (Fig. 3d)

## (b) Volcanism

Smoke is burning/rising from Satsuma Iwojima
• Nubui Kuduchi verse 8

Kaimondake comes into view and Sakurajima's majesty could be mistaken for Mount Fuji
• Nubui Kuduchi verse 8

### Satsuma-Iwojima

Young ages of Iwodake lava (Kawanabe and Saito, 2002) and observed 20th–21st century fumaroles and ash eruptions (Kamada, 1964; Kazahaya et al., 2002; Shinohara et al., 2002; Fig. 4b).

### Sakurajima and Kaimondake

The unincised Kaimondake edifice (Fujino and Kobayashi, 1997) is more geomorphically similar to Mount Fuji than Sakurajima (Fig. 4c–e).

Lack of historical recorded eruptions during the time of the envoy precludes that this mismatch is due to a catastrophic Sakurajima eruption (Fig. 4d–e; Kobayashi and Tameike, 2002).

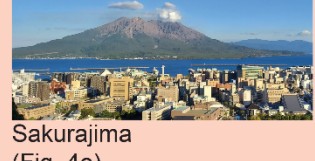 Sakurajima (Fig. 4e)

294

**Figure 5: Linking lyrical observations with scientific literature on climate and volcanic systems that Ryukyuan envoys may have experienced. First, we extract (a) climate and (b) volcanic observations from Nubui Kuduchi and Kudai Kuduchi (top). Then, we subgroup lyrics that correspond with scientific literature on the East Asian Monsoon, ocean conditions, Satsuma-Iwojima, or Sakurajima and Kaimondake, related to oceanic currents (Fig. 1), landmarks (Fig. 3), and volcanoes (Fig. 4) observable or visitable in the 21st century (bottom).**

### 3.2.2 Lyrical links to climate

We link the south-southwesterly winds described in Nubui Kuduchi with 20th–21st-century observations of the south to southwesterly winds that prevail during the Boreal summer monsoon from May to September (Dobby, 1945; Flohn, 1957; Fu et al., 1983; Ueda et al., 1995). As Nubui Kuduchi does not explicitly mention the timing of Satsuma-bound envoys, we utilize historical records and genealogies that show the ~20 departures from Naha port occurred between the fifth and eighth months of the lunar calendar (approximately May to August; Okinawa Prefectural Cultural Promotion Association, 2001; Toby, 1984), agreeing with lyrics and the scientific literature (Fig. 5a). After arriving in Satsuma and traveling to and from Edo, Kudai Kuduchi describes the envoys returning to Okinawa Island on early autumn north-northeasterly winds. Historical records show that envoys returned to Naha port between the eleventh and fourth month of the lunar calendar (approximately November to April; Okinawa Prefectural Cultural Promotion Association, 2001), consistent with the northeasterly Boreal winter monsoon from September to May (Dobby, 1945; Fu et al., 1983). Accordingly, we can link the specific directions and seasonality of winds in Nubui Kuduchi, Kudai Kuduchi, and historical records with the East Asian monsoon system in scientific literature.

Both songs then mention prayers for safe travels, implying a dangerous voyage. We link this and other lyrical concerns for ocean conditions with three scientific points (Fig. 5a). First, the envoy intersects the apex of the Kuroshio current at the famously dangerous seas of the Tokara Strait (Fig. 1). This region is associated with high wave heights from ocean-atmosphere coupling (Hwang, 2005) and interactions with seamount bathymetry that induce turbulence in the near-surface ocean (Nagai et al., 2021; Tsutsumi et al., 2017). Second, Holocene geologic evidence, modern climatological data, and numerical models highlight the commonality of typhoons that cross the envoy's approximate path, particularly during El Niño periods of ENSO (Ito et al., 2020; Sun et al., 2017; Wang and Chan, 2002; Wu and Wang, 2004; Yang et al., 2020). In fact, the timings of some journeys put certain envoys in the timeframe of the highest typhoon activity from July to September (Wu and Wang, 2004). Third, the region between Kyushu and Taiwan has relatively high average and maximum wave heights compared to surrounding seas, in part due to the presence of these typhoons (Wu et al., 2014; Young et al., 2011). Therefore, Nubui Kuduchi and Kudai Kuduchi can be linked with the impacts of ocean currents, typhoons, and ocean-atmosphere coupling on dangerous ocean conditions in the western Pacific Ocean.

## 3.3 Volcanism

### 3.3.1 Ryukyu volcanism review

Philippine Sea Plate subduction under the Eurasian Plate occurs east of the Ryukyu Islands at the Ryukyu Trench (Fig. 1; e.g., Kamata and Kodama, 1994). Quaternary island arc volcanism, characteristic of such ocean-continent subduction, is focused north of Okinawa Island (Global Volcanism Program, 2024). In particular, the northern Ryukyu region includes Kikai Caldera, which produced at least three ignimbrite eruptions including one in the mid-Holocene with a volcanic explosivity index of seven out of eight (Fig. 4a; Machida and Arai, 2003; Maeno and Taniguchi, 2007; Ono et al., 1982).

Satsuma-Iwojima ("iwo" referring to $i\bar{o}$, 硫黄, or sulfur in Japanese) is an island on the perimeter of this submarine caldera composed of two volcanoes with different lava types: the basaltic Inamuradake cone and the larger, rhyolitic Iwodake dome (Fig. 4b). The oldest recorded observation of Iwodake activity is from Japanese written tradition, the Heike Monogatari (平家物語), which details late-12th-century volcanism, sulfur mining (Kamada, 1964; Kazahaya et al., 2002; Shinohara et al., 2002), and political banishments to the island (Antoni, 1991). However, subsequent records of volcanic activity from this period are sparse despite a likely Iwodake eruption between ~1300–1450 CE, determined from calibrated [14]C radiometric ages in decimeter-thick rhyolite flows (Kawanabe and Saito, 2002). Residents also reported active fumaroles (volcanic gas vents) and small eruptions but with no recorded timings or further details (Kamada, 1964; Shinohara et al., 2002). Nonetheless, Iwodake gas discharge events were recorded in the 20th and 21st centuries, producing transient, fine, ash fall deposits that were observable only on smooth surfaces up to ten kilometers away (Kazahaya et al., 2002; Shinohara et al., 2002). Research has attributed this activity to Kikai Caldera magma and degassing, highlighting an interconnected magma conduit at depth (Saito et al., 2001) and activity for over 1000 years (Kawanabe and Saito, 2002; Shinohara et al., 2002). In the 21st century, such activity has extensive impacts on water quality by hydrothermal leaching (e.g., Kiyokawa et al., 2012) and air quality by sulfate aerosols (e.g., Itahashi et al., 2019) locally and throughout the western Pacific region.

Extensive caldera complexes are also located in southern Kyushu (Nagaoka, 1988). The Ata and Aira Calderas host the Quaternary-active stratovolcanoes Kaimondake and Sakurajima, respectively (Fig. 4a). Kaimondake is a relatively uneroded cone at the entrance to Kagoshima Bay that last erupted in 885 CE (Fig. 4c; Fujino and Kobayashi, 1997). Roughly 50 km north, Sakurajima is a volcano in Kagoshima Bay and hosts three peaks: Kitadake, Nakadake, and Minamidake (translated as north, central, and south peaks, respectively; Fig. 4d–e). In particular, Minamidake and surrounding fissures have four recorded major eruptions between the 8th–20th centuries, impacting communities including the city of Kagoshima less than ten kilometers away (Fig. 4a; Kobayashi and Tameike, 2002). Sakurajima remains active in the 21st century, with nearly 3000 Vulcanian-style eruptions between 2008–2011 (e.g., Iguchi et al., 2013).

### 3.3.2 Lyrical links to volcanism

Upon arrival in Satsuma, Nubui Kuduchi mentions Kaimondake and Sakurajima but likely only holds geologically interpretable observations of Satsuma-Iwojima (see Sect. 4.1 for Kaimondake and Sakurajima). Here, the song describes Satsuma-Iwojima activity as a gas or ash plume. We link these lyrical observations with the historical (Kamada, 1964) and scientific (Kazahaya et al., 2002; Shinohara et al., 2002) observations of the active Iwodake fumaroles and ash eruptions documented in the 20th–21st centuries (Fig. 5b). In fact, lyrics of burning or glowing smoke are similar to the discharge of hot and illuminated gases from a 1996 nighttime eruption described in Shinohara et al. (2002). Therefore, 18th-century Nubui Kuduchi adds an intermediary observation of Iwodake activity between the 12th-century Heike Monogatari and 20th–21st-century scientific studies. We exemplify how Indigenous knowledge from Nubui Kuduchi supports relatively consistent

volcanic activity from Iwodake as suggested by radiometric dating and observed eruptions (Kawanabe and Saito, 2002;
Shinohara et al., 2002).

## 4 Discussion

### 4.1 Lyrical implications for historical climate and volcanism

The similarity of seasonal wind directions in Nubui Kuduchi and Kudai Kuduchi lyrics, historical documents
(Okinawa Prefectural Cultural Promotion Association, 2001; Toby, 1984), and the modern monsoon (Dobby, 1945; Flohn,
1957; Fu et al., 1983; Ueda et al., 1995) implies a reliance on predictable monsoonal winds for the ~20 Ryukyuan envoys
going to and from Satsuma (Okinawa Prefectural Cultural Promotion Association, 2001; Toby, 1984). Deviations from this
norm could be highlighted and explored as monsoonal abnormalities. For example, genealogies suggest that a 1791 CE
envoy was required to wait in Satsuma between the fourth to tenth lunar months of 1791 CE due to unfavorable winds earlier
in the year (Okinawa Prefectural Cultural Promotion Association, 2001). These records could signify a failure of the East
Asian Boreal summer monsoon that coincided with El Niños between 1788–1796 CE, known collectively as the Great El
Niño (Cook et al., 2010; Grove, 2006; Quinn et al., 1987). Records show that envoys did not resume until 1796–1797 CE
near the end of this climate anomaly (Okinawa Prefectural Cultural Promotion Association, 2001; Toby, 1984), but more
historical analyses are required to assert that this envoy gap reflects a monsoon failure. Nubui Kuduchi and Kudai Kuduchi
then detail prayers and tears shed for the dangerous journey ahead (e.g., at locations in Fig. 3a–b), foreshadowing deadly
typhoons, waves, and ocean currents. Speculatively, Ryukyuan voyagers could have been indirectly praying for La Niña
conditions associated with a lower probability that typhoons will track toward the Ryukyu Islands (Ito et al., 2020; Sun et al.,
2015, 2017; Wang and Chan, 2002; Wu and Wang, 2004; Yang et al., 2020). Likewise, the envoy may have indirectly
recorded the impacts of typhoons (Wu et al., 2014; Young et al., 2011) and the Kuroshio Current (Hwang, 2005; Nagai et al.,
2021; Tsutsumi et al., 2017) on ocean conditions in and around the Tokara Strait, which is highlighted as a treacherous
location in both songs. As such, Nubui Kuduchi and Kudai Kuduchi provide a proof-of-concept for extracting historical
climate from RKO to connect the cultural arts with geoscience in the Ryukyu Islands.
Voyages north of Okinawa Island, where active volcanoes are located (Fig. 1), then offer a rare opportunity to
record volcanism in Ryukyuan songs. If Nubui Kuduchi describes average degassing across the attributed composer's
lifetime (1716–1775 CE; Gillan, 2012; Kinjo, 1992) or the envoy's activity (1610–1872 CE; Okinawa Prefectural Cultural
Promotion Association, 2001; Toby, 1984), perhaps this song represents volcanic processes over ~60 or ~250 years,
respectively. Regardless, this volcanic benchmark from 18[th]-century Nubui Kuduchi bridges an ~800-year gap between the
12[th]-century Heike Monogatari and 20[th]–21[st]-century studies at Satsuma-Iwojima and Kikai Caldera (e.g., Kawanabe and
Saito, 2002; Kazahaya et al., 2002; Saito et al., 2001; Shinohara et al., 2002). Such a gap may be in part because Satsuma-
Iwojima's population was historically ignored and othered by Japanese society (Antoni, 1991). Thus, lyrical observations
from the Ryukyu Kingdom can help scientists understand the continuity of volcanic activity on this island. For example, the
long-term degassing suggested by this song agrees with evidence of over 1000 years of eruptive activity from
radiometrically dated Iwodake rocks (Kawanabe and Saito, 2002; Shinohara et al., 2002). These observations fit into a
previously proposed model of a long-term, convecting, and stratified Satsuma-Iwojima magma chamber that feeds basaltic
Inamuradake and rhyolitic Iwodake. Workers suggest that denser basalt magma may sit lower in the chamber to 1) supply
the less dense rhyolite magma above with volatile gases that 2) cause this rhyolite to ascend in the Iwodake magma column
and 3) induce surface degassing by decompression to 4) cause this magma to descend once degassed of buoyant volatiles
(Kazahaya et al., 2002; Saito et al., 2001; Shinohara et al., 2002). Therefore, Nubui Kuduchi may connect surface
phenomena with supposedly long-lived and deep magmatic processes at Satsuma-Iwojima over ~800 years.
Nubui Kuduchi also compares Sakurajima's geomorphology to that of Mount Fuji. However, the uneroded and
single-peaked Kaimondake (Fig. 4c) is more similar to the conical Mount Fuji than Sakurajima (Fig. 4d; Fujino and
Kobayashi, 1997). Historical records from this well-populated area preclude that this discrepancy signals an eruption that
dramatically altered Sakurajima's shape to its current form (Kobayashi and Tameike, 2002). The lyrics are likely a poetic
interpretation of the first sight of this volcano. Alternatively, the north-south alignment of summit peaks and the south-
opening Kagoshima Bay suggest that arriving voyagers would see a more conical volcanic profile than if seen from the east
or west (Fig. 4e–f). This geomorphology may also lead to the generous interpretation of Sakurajima as mistakable for Mount
Fuji.
**4.2 Future of place-based *Ryūkyū koten ongaku***
The links we establish between Nubui Kuduchi, Kudai Kuduchi, and scientific literature can be utilized in place-
based engagement to highlight the impacts of climate and volcanism on 21[st]-century Okinawan students. For Okinawans in
Okinawa Prefecture, students may examine the East Asian monsoon system, ENSO, and observable wind patterns on which
18[th]-century voyagers likely relied (following Dobby, 1945; Flohn, 1957; Fu et al., 1983; Ueda et al., 1995). Likewise, the
impacts of typhoons and the Kuroshio Current hinted at in both songs provide ancestral connections to natural disaster
awareness (following Ikema et al., 2010) and oceanic voyaging in Okinawa. Visiting the sites where voyagers acted upon
their knowledge of dangerous wind and ocean conditions, which are now famous landmarks (e.g., Fig. 3), could emphasize
place-based links between Indigenous knowledge and scientific literature (Fig. 5; Semken et al., 2017). Farther outboard,
Nubui Kuduchi presents an opportunity to teach recent Ryukyuan volcanology and the complex geologic processes of the
Ryukyu Trench and Kikai Caldera (following Kamada, 1964; Kazahaya et al., 2002; Shinohara et al., 2002). Critical
thinking modules can then allow students to hypothesize how climate change may modify Ryukyuan weather and storms
(e.g., Kitoh, 2017; Lu et al., 2007) or examine how modern scientists quantify volcanic impacts on air and water quality
(e.g., Itahashi et al., 2019; Kiyokawa et al., 2012). For the Okinawan diaspora, modules on Ryukyuan climate change or
volcanic hazards could be showcased at popular Okinawan events, such as HUOA's Okinawan Festival. Place-based
geoscience connections could also be incorporated into field trips when visiting Okinawa Prefecture for the Worldwide
Uchinanchu Festival (Okamura, 2022) or *kahi* tours (Gillan, 2017). Thus, educators can engage 21[st]-century learners in
Okinawa Prefecture and across the overseas diaspora using the links to contemporary science that we have found in Nubui
Kuduchi and Kudai Kuduchi.

430         Similar geoscience insights could be examined from additional Ryukyuan performing arts. Following the works on

Indigenous knowledge of freshwater resources in Takahashi (2022), Nuchibana (貫花) and Amakā (天川) are popular songs
that also speak of rivers and water around Okinawa (Seki, 2024). Then, descriptions of the Okinawan nearshore environment
in Toguchi and Nishime (2013) and Toguchi et al. (2016) parallel songs such as Tanchamē (谷茶前) and Umi nu Chinbōrā
(海ぬチンボーラー) that make observations of shallow sardine and cone snail species, respectively (Seki, 2024). These
commonly performed numbers (Hanashiro, 2007) may highlight local hydrological, biological, and geological phenomena to
familiarize Okinawans with their environment. Therefore, future work in preserving and compiling Indigenous knowledge
within the Ryukyuan arts could be key for place-based engagement across the Okinawan community.
**4.3 Empowering Okinawan communities**

439         Incorporating Nubui Kuduchi and Kudai Kuduchi in geoscience engagement can uplift Ryukyuan culture, language,

and indigeneity against the backdrop of historical marginalization. Okinawan diasporas have a unique opportunity to connect
with other Indigenous Peoples who have had successful experiences with place-based geoscience education (e.g., Hawaiʻi;
Chinn et al., 2014; Gibson and Puniwai, 2006). In fact, Native Hawaiian revitalization and sovereignty movements have
directly influenced parallel efforts to revitalize Okinawan culture and languages (e.g., Heinrich, 2018; Kina, 2020).
Conversations between the highly active Hawaiʻi-Okinawa diaspora, Okinawans in Okinawa Prefecture, and the Native
Hawaiian community have facilitated cross-cultural exchange towards Okinawan cultural resurgence (e.g., LooChoo Identity
Summit, where LooChoo is an approximation for one Ryukyuan language's pronunciation of Ryukyu; Ohara and Slevin,
2019). Further collaborations in place-based pedagogy within these preexisting relationships may help legitimize Ryukyuan
Indigenous knowledge and grow this Indigenous movement.

449         It is important for this and any future RKO works to empower Indigenous practitioners, ensure rightful recognition,

and present accurate interpretations. For example, many Ryukyuan songs that describe nature are steeped in metaphor,
including human relationships (e.g., Karaya Bushi; 瓦屋節), political rebellion (e.g., Unna Bushi; 恩納節), and
homesickness (e.g., Chijuyā; 浜千鳥; Sakiyama and Oshiro, 1995; Seki, 2024). These factors often make scientific
interpretations difficult (e.g., Swanson, 2008), but can be overcome by continued collaborations with Indigenous elders and
scholars similar to how J.Y. Uyeunten and K.A. Odo guided our interpretations of Nubui Kuduchi and Kudai Kuduchi
(Younging, 2018). Although it is challenging to compare poetic observations from songs with scientific observations from
literature, the ability to experience the same environmental conditions as sung therein can enhance hands-on and place-based
science engagement in the 21[st] century. Future collaborations with RKO leaders are required to interpret additional songs and
develop engagement modules for this purpose (Sect. 4.2). As such, there is great potential for more partnerships between the
Ryukyuan arts and geoscience engagement.

## 5 Conclusions

RKO contains historical observations of the natural world that describe atmospheric, oceanic, and geologic processes. Here, we demonstrate how Nubui Kuduchi and Kudai Kuduchi hold 18th-century descriptions of winds, currents, typhoons, and volcanoes in the Ryukyu Islands of the western Pacific. Through a novel collaboration with cultural practitioners, we show that observations of the natural world in lyrics correspond with climate and volcanological research in 20th–21st-century scientific literature. Such correspondence suggests that Indigenous knowledge in these songs can be used to better engage Okinawans in Okinawa Prefecture and across the global Okinawan diaspora with complex geoscience topics applicable to their communities. Educators can apply these lessons to place-based science modules centered on the specific climate, geologic, and social issues facing 21st-century Okinawans. Thus, RKO is pedagogically vital for promoting science engagement in and about the Ryukyu Islands.

## Data availability

Data produced in this project consist of two video supplements for Nubui Kuduchi and Kudai Kuduchi (Higa et al., 2024a, b).

## Video supplement

See video supplement for documentation of Nubui Kuduchi and Kudai Kuduchi lyrics, translations, and interpretations (Higa et al., 2024a, b).

## Author contribution

Conceptualization: JTH, Data curation: JTH, Formal analysis: JTH, JYU, KAO, Funding acquisition: JTH, Investigation: JTH, JYU, KAO, Methodology: JTH, JYU, KAO, Project administration: JTH, Resources: JTH, JYU, KAO, Supervision: JTH, Validation: JTH, JYU, KAO, Visualization: JTH, Writing–original draft preparation: JTH, Writing–review & editing: JTH, JYU, KAO.

## Competing interests

The authors declare that they have no conflict of interest.

## Ethical statement

All authors are members of the Ryukyu Koten Afuso Ryu Ongaku Kenkyu Choichi Kai USA, Hawai'i Chapter. KAO and JYU are Master Instructors within this organization and permit the publication of these works with their expert discussion. JTH has obtained verbal permission from the founder of this organization, G.S. Murata, to continue with this work and from leaders of the Hooge Ryu Hana Nuuzi no Kai Nakasone Dance Academy, J. Okamura and L. Nakandakari, to adapt translations and interpretations from C.T. Nakasone. All supplemental data are performed by the authors of this paper, who approve and are informed of the contents therein.

## Acknowledgements

We thank M.O. Argueta, S.J. Coats, J.H. Hewitt, S.K. Izuka, L. Nakandakari, K.E. Odo, J. Okamura, and M.G. Robbins for discussion, K. Ikei, K.E. Odo, and S. Tomori for literature and materials, T. Irei and K. Sakihara from the Okinawa Prefectural Museum and Art Museum for references, B. Kuhasubpasin for Fig. 2 illustrations, and two anonymous reviewers and *Geoscience Communication* editors for constructive reviews. We highlight the founder of the Ryukyu Koten Afuso Ryu Ongaku Kenkyu Choichi Kai USA, Master Instructor G.S. Murata, for support. We are also grateful to the Hooge Ryu Hana Nuuzi no Kai Nakasone Dance Academy and Master Instructors J. Okamura and L. Nakandakari for their permission to adapt translations and interpretations by the late C.T. Nakasone.

## Financial support

This material is based upon work supported by the National Science Foundation under Award Number 2305448.

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
