# Peer review of "Place-based science from Okinawa: 18th-century climate and geology recorded in Ryukyuan classical music"

_EGUsphere, 2025_

## Author Comment (AC1)

*Dear Reviewer,*

*Thank you very much for your highly constructive feedback. Below, we respond to your comments in blue italics. We hope these responses adequately address your efforts to improve the scientific rigor of our manuscript. We will provide our textural changes once all reviews are received.*

An intriguing paper that describes how Ryukyuan classical music could play an important role in engaging Okinawans and those in the Okinawan diaspora in the geosciences and science education.  I enjoyed listening to the examples of Nubui Kuduchi and Kudai Kuduchi; this is definitely a strength of this paper.

*We thank you for listening to our recording and appreciating its contribution to our manuscript.*

While this paper has the potential to contribute to scientific progress within the scope of Geoscience Communication, it fails to do so due to its lack of claim for significance (*so what?*) that is solidly grounded in the scholarly literature.

*We hope our responses to your suggestions below will address our claim to significance by better situating ourselves in the Indigenous knowledge and science education literature, particularly in geoscience works. To summarize our edits:*

1. *We improve the organization of our manuscript. Mainly, we move the background information on the Okinawan people and diaspora from the Discussion to the Introduction sections. We also focus on historical discrimination and present activity in Okinawan culture to highlight why our work is important for geoscience communication. We then split up several long in-text citations to better explain the significance of each work.*

2. *We better present the knowledge gap our work aims to cover, particularly that Indigenous music can be used to interpret geoscience topics and engage Indigenous communities. We highlight the gap in research on Ryukyuan classical music (Ryūkyū koten ongaku; RKO) regarding geoscience, which, to our best knowledge, has never been done before in English or Japanese literature.*

3. *We expand on our methods section by splitting it into how we 1) performed our lyrical synopses in collaboration with Indigenous practitioners and 2) performed our literature review with a specific focus on observations of climate and volcanic systems. Due to this change, we include our Background/literature review as a result in our new Results section.*

From what is presented, the paper has the potential to contribute to (a) the literature on Indigenous Knowledge and how traditional music can help scientists better understand historical as well as more recent climatic and geological processes; and (b) the literature on how traditional music, especially those of marginalised communities, could be used to improve

engagement of these communities with science, and empower them.  Unfortunately, the authors fail situate the article well within the scholarly literature.  In fact, the literature review in the Introduction section is too superficial for the reader to get a good understanding of the literature.  As for (a), it is unclear what the literature discusses on how traditional music can help scientists better understand historical as well as more recent climatic and geological processes.

*We greatly appreciate your detailed, constructive criticism of our Introduction here.*

*To address (a), we rewrite lines 21-24 to explicitly state how the cited works have contributed to including Indigenous knowledge in studying climate science or geologic hazards, not necessarily music (e.g., sea-level rise, earthquakes, volcanoes). To situate our study in traditional music, we then cite the success of works that have used Indigenous performing and visual arts to scientifically analyze geologic and environmental phenomena (e.g., Hough, 2007; Ludwin et al., 2005; Swanson, 2008). Furthermore, we cite new literature that exemplifies how incorporating multiple ways of knowing (such as that recorded in Indigenous art) is beneficial for science by increasing research objectivity and creativity (Bang et al., 2018; Am. Acad. Arts Sci.; Intemann, 2009; Social Epistemology).*

As for (b), four articles are cited as highlighting "previous works highlight how incorporating indigenous stewardship in Earth science courses can increase engagement by emphasizing the local, land-human interactions to which a learner has an ancestral connection" but we have no idea what these previous works have said on this topic.  Thus, it is unclear to the reader how this paper contributes to the literature on (a) and (b).

*We acknowledge the lack of background from the cited papers in this section. To accommodate point (b), we specifically introduce the sources of indigenous knowledge used by Chinn et al. (2014) and Gibson and Puniwai (2006) to increase geoscience engagement in Hawai'i. We also describe the positive outcomes of including these multiple ways of knowing described in Alexiades et al. (2021). We then add Palmer et al. (2009; Journal of Geoscience Education), that utilized North American Indigenous art in an undergraduate course to showcase how art can increase geoscience engagement, similar to our work here.*

Similarly, it is unclear from the brief mentions of the literature at the end of p.2 to the first paragraph on p.3, what has been written so far (if any) on the topic of Ryukyuan classical music and climatic/geological processes.  Is this paper filling a gap in any literature?  If so, what exactly are the gaps?

*Thank you for pointing out this deficit in our manuscript. We now discuss the background behind current work that examines Ryukyuan indigenous knowledge on scarce island groundwater (Takahashi, 2022), regional fisheries (Toguchi et al., 2016), and coral reef geobiology (Toguchi and Nishime, 2013). We also explicitly state that there is a gap in researching Ryukyuan classical music in terms of geoscience engagement in both English and Japanese literature (to our best knowledge) despite Ryukyuan classical music being a very visible component of Okinawan culture today.*

The methods is another section where scientific rigour could be improved; the methods are not described comprehensively. In particular, it would be good for the reader to know (1) what are the other versions/schools of Nubui Kuduchi and Kudai Kuduchi exist, why the version used for analysis in the paper was used (is it the "mainstream" version? Why was it not compared with other versions/schools?);

*Thank you for this constructive comment towards improving our Methods section.*

*To address point (1), we now add sentences on why we opted to use the version of the songs here. As members of the mainstream Afuso Ryu school of Ryukyuan classical music, our lineage only covers the versions that we present here. However, J.Y. Uyeunten and K.A. Odo are also knowledgeable about the one other mainstream school called Nomura Ryu. The differences between schools are mainly in how the song is executed through singing and playing; there are only a few differences in lyrics, and the most relevant difference regarding volcanoes is already noted in the Results section. This overall similarity is likely because both schools diverged from the same lineage relatively recently in the 1800s. Gillan (2012) and new citation Garfias (1993; Asian Music) discuss how Afuso Ryu preserved the style of the original lineage while Nomura Ryu simplified the style to make the art more accessible. Therefore, we add sentences to explicitly discuss such version control, differences between versions, and why we use our version.*

(2) a description of the Okinawan diaspora in general, and the diaspora in Hawai'i, …

*To address point (2), instead of in the Methods, we expand our newly added Introduction Section 1.1 (described in-depth in a later response) to describe the Okinawan community in Okinawa Prefecture, in the diaspora, and the differences between both groups. We then add a review on where diasporas are located, the number of immigrants, what kind of labor attracted them to go overseas, discrimination faced overseas, and the distinct Okinawan identity retained through 4+ generations, with a focus on the one in Hawai'i (e.g., population, affinity groups, cultural events).*

… and what kind of differences would the reader need to be aware of, in terms of the Okinawans in Okinawa vs. the diaspora elsewhere (also related to (1) above, are there differences in Nubui Kuduchi and Kudai Kuduchi sung by Okinawans in Okinawa and those transmitted within the diasporas?);

*We then add a review on the differences between Okinawans in Okinawa and abroad: the language and culture barrier after separation from Okinawa by 4+ generations. We also address how Ryukyuan classical music instructors in the diaspora are often trained in Okinawa or trained by instructors from Okinawa. This training ensures that the versions transmitted in the diaspora are the same as those sung in Okinawa Prefecture.*

(3) more details on methodology used to "link" observations of climatic, geological, and environmental processes, as the authors refer to Swanson's work. This may help the reader understand better how photos and images (Figures 2, 4) were used in the analysis.

*To address point (3), we expand on and organize our Methods Section 2 into two subsections: 1) the Indigenous sources of lyrics and how we created our lyrical synopses and 2) the specifics of how we performed our literature review to find the concurrence between song and science.*

*In new Section 2.1, we include our background of Nubui and Kudai Kuduchi and the various sources of Ryukyuan indigenous knowledge and modern literature used to supplement our synopses. This is where we discuss the various permissions we obtained from Ryukyuan arts practitioners, how we collaborate between art and science practitioners to ensure rightful recognition of Ryukyuan sources, and why we base our interpretation off one version of both songs (i.e., the lyrics we are interpreting are very similar across the two main Ryukyuan classical music schools, described above).*

*In new Section 2.2, we provide more details on our literature review to find the correspondence or "links" we found between lyrics and the literature. Here, we describe the methodology of Swanson (2008), namely utilizing one source of a Hawaiian oral history/chant/dance, of which there may be many versions, to compare with contemporary scientific literature. We note that this situation is similar to ours, where there is more than one version of both songs, but the meanings are fundamentally the same. We summarize that Swanson (2008) finds correspondence between Western and Indigenous ways of knowing.*

What is written in sections 4.2-4.3 show that the lyrics and historical records concur; how was this information analysed to lead to the discussion in section 5.1?

*In Section 2.2, we then describe how we applied the framework of Swanson (2008) and focused on a literature review that specifically aims to find 1) similarities or differences between observations in Nubui/Kudai Kuduchi and modern literature (e.g., concurring monsoonal winds and volcanic activity) and 2) how the climate and volcanism described in song and science can affect humans today (e.g., impacts of climate change on typhoons, air pollution from volcanic activity). Discussion Section 4.1 (previous Section 5.1) thus relates to addressing point 1) and how the concurrence between Ryukyuan observations in song and modern instrumented science can lead to interesting and novel scientific inferences. Section 4.2 (previous Section 5.2) will relate to addressing point 2) by using these scientific links to connect environmental science issues in Okinawa with students through place-based learning.*

*Following these changes, we reorganize the Results Section 3, where 1) the literature review constitutes a separate "result" followed by 2) the correspondence in climate or volcanic*

*processes that we propose are found in song and science. Thus, the literature review work does not precede the methods that describe how and why the literature review was done.*

*Moreover, we add another figure showing a diagram of how specific verses are related to literature in our review. We cite reordered Figures 1, 3 (originally Figure 4), and 4 (originally Figure 2) to emphasize how the observable phenomena or visitable sites in those figures are related to each link.*

[Figure]

*Figure #: Summarized 18th-century verses of Nubui Kuduchi (left) and Kudai Kuduchi (right) linked with 20th-21st-century scientific observations and literature on climate and volcanism (center). Climate and volcanism are divided into specific phenomena or locations that are likely sung about in both songs. Some verses or observations may discuss oceanic conditions (Fig. 1), landmarks (Fig. 3), and volcanic landforms (Fig. 4) that are observable or visitable in the 21st century.*

Sections 5.2 and 5.3 appear to be literature review sections, much of which could be brought to the Introduction section to show us the gaps (or, what this paper builds on) in the literature. Once again, the authors would need to include more details so the reader can get a better understanding of the literature on the topic.

*Thank you for pointing out this needed reorganization. We have moved much of the original Discussion sections to the Introduction to highlight the gaps and demand for our work in Ryukyuan indigenous knowledge. We also expand upon our discussion of the Ryukyu Kingdom, Ryukyu colonization and discrimination, the Okinawan diaspora, Ryukyuan classical music, and the environmental themes likely recorded in Ryukyuan classical music that could be useful for geoscience engagement.*

*In Section 1, we move up the previous work in Ryukyuan Indigenous knowledge and geoscience (i.e., moved the second paragraph in the original Section 5.2 and citations Takahashi, 2022; Toguchi et al., 2016; Toguchi and Nishime, 2013) and the gap in utilizing Ryukyuan classical music in geoscience, as discussed in a previous response.*

*In the new Section 1.1, we provide more details on the Ryukyu Kingdom, its conquest and annexation by Japan, assimilation policies implemented by Japan, and post-World War II history that led to revitalization efforts of Ryukyuan culture (i.e., moved much of the original Section 5.3 on Empowering Okinawan Communities to the Introduction).*

*Then, we include more details on the Okinawan diaspora as highly active and distinct from the Japanese diaspora. This Okinawan diaspora is another audience for the work done here due to historical discrimination and a demand to reconnect with their Okinawan heritage.*

*We also provide more details on Ryukyuan classical music. Specifically, we include how Ryukyuan classical music is still relevant in Okinawa today due to the support of classical music schools, instructors, and organizations, as well as prefectural government support.*

I also recommend that the authors state the argument of the article clearly in the introduction. Currently, the first time the argument appears (after the abstract) is at the end of the methods section. It would be helpful for the reader to see this earlier.

*Thank you for pointing this out. We will more clearly state the argument of our paper in the Introduction.*

Once again, the fact that the authors have experiences and expertise as Ryukyu classical musicians makes this paper particularly appealing. Once the scientific rigour is strengthened, the paper could make an important contribution to scientific progress within the scope of Geoscience Communication.

*Thank you again for your helpful and constructive feedback for improving the scientific rigor of our manuscript.*

---

## Author Response (AR1)

*Dear Editor and Reviewers,*

*Thank you for all your thorough, thoughtful, and constructive reviews of this manuscript. Below, we provide our responses (in blue italics) for your comments (in black roman type). We expand upon our updated responses from the public review by showing our quoted text changes based on those suggestions. The line numbers provided for quotes refer to the revised manuscript PDF with tracked changes turned off or the manuscript file with no tracked changes. Full quotes are provided for your convenience but are also available in the tracked changes and manuscript documents; we do not provide some quotes for paragraphs that were simply moved to a new location with minor corrections but do provide line numbers for these passages. We look forward to hearing your responses.*
* * *
*Dear Editor*

*Thank you for your support and summary of our reviews. Please see our responses to them below.*

Thank you for a very constructive and contrastive discussion between the reviewers and the authors. The authors have clearly considered all reviewer comments and suggestions and responded thoughtfully. Here are my responses to the two discussions:

RC1:
The most pertinent point raised by the reviewer concerns situating the paper within the scholarly literature and clarifying the so what of the research. I fully support the reviewer's recommendation that the authors clearly state the aim of the article (and preferably a specific research question) in the Introduction. I am satisfied with the way the authors have addressed all the reviewer's comments. However, given the substantial nature of the reviewer's feedback, I recommend another read-through by the reviewer before the final version is approved for publication. This is particularly important as a considerable amount of new text—and a new figure—has been added.

*Thank you for your suggestion to rephrase our "so what" of this paper as a specific research question. Please see below our revised research question and surrounding context.*

> *Line 48 - 55: "However, to the best of our knowledge, no works in English or Japanese have examined the connections between geoscience and one of the most influential Indigenous art forms across Okinawa and the Okinawan diaspora, Ryukyuan classical music, also known as Ryūkyū koten ongaku (琉球古典音楽; hereafter RKO). We compare this tradition with contemporary science across the atmo-, hydro-, and geosphere to address the question: Does RKO record place-based environmental phenomena useful for*

*geoscience engagement? Here, we focus on a popular RKO repertoire, Nubui Kuduchi (上り口説) and Kudai Kuduchi ( 下り口説), to trace the 18th-century experiences of Ryukyuan seafaring envoys between Okinawa and Kyushu islands (Okinawa Prefectural Cultural Promotion Association, 2001; Toby, 1984)."*

*As in our response to RC1, we end this paragraph stating our argument (i.e., the answer to the above research question) more clearly.*

> *Line 60 - 62: "Accordingly, we showcase how popular Indigenous music can document scientific observations of climate and geology to engage Indigenous Peoples with their contemporary environmental heritage."*

RC2:
The comments from RC2 are less substantial than those from RC1, but still very useful and constructive. I am satisfied with the way the authors have responded.

*Please see below our full revisions based on RC2's comments, including revised figures based on their useful suggestions.*

Based on the discussion and the changes proposed by the authors, I am happy to recommend publishing the article—pending final approval from the reviewers after the revisions have been reviewed.

*We greatly appreciate your work handling our manuscript and hope our text changes in response to your and reviewer comments have adequately addressed all concerns.*

*Dear Reviewer 1,*

*Thank you very much for your highly constructive feedback. We hope these responses adequately address your efforts to improve the scientific rigor of our manuscript.*

An intriguing paper that describes how Ryukyuan classical music could play an important role in engaging Okinawans and those in the Okinawan diaspora in the geosciences and science education. I enjoyed listening to the examples of Nubui Kuduchi and Kudai Kuduchi; this is definitely a strength of this paper.

*We thank you for listening to our recording and appreciating its contribution to our manuscript.*

While this paper has the potential to contribute to scientific progress within the scope of Geoscience Communication, it fails to do so due to its lack of claim for significance (*so what?*) that is solidly grounded in the scholarly literature.

*We hope our responses to your suggestions below will address our claim to significance by better situating ourselves in the Indigenous knowledge and science education literature, particularly in geoscience works. To summarize our edits:*

1. *We improve the organization of our manuscript. Mainly, we move the background information on the Okinawan people and diaspora from the Discussion to the Introduction sections. We also focus on historical discrimination and present activity in Okinawan culture to highlight why our work is important for geoscience communication. We then split up several long in-text citations to better explain the significance of each work.*

2. *We better present the knowledge gap our work aims to cover, particularly that Indigenous music can be used to interpret geoscience topics and engage Indigenous communities. We highlight the gap in research on Ryukyuan classical music (Ryūkyū koten ongaku;* **which we now abbreviate as RKO**) *regarding geoscience, which, to the best of our knowledge, has never been done before in English or Japanese literature.*

3. *We expand on our methods section by splitting it into how we 1) performed our lyrical synopses in collaboration with Indigenous practitioners and 2) performed our literature review with a specific focus on observations of climate and volcanic systems. Due to this change, we include our Background/literature review as a result in our new Results section.*

From what is presented, the paper has the potential to contribute to (a) the literature on Indigenous Knowledge and how traditional music can help scientists better understand historical as well as more recent climatic and geological processes; and (b) the literature on how traditional music, especially those of marginalised communities, could be used to improve engagement of these communities with science, and empower them. Unfortunately, the

authors fail situate the article well within the scholarly literature.  In fact, the literature review in the Introduction section is too superficial for the reader to get a good understanding of the literature.  As for (a), it is unclear what the literature discusses on how traditional music can help scientists better understand historical as well as more recent climatic and geological processes.

*We greatly appreciate your detailed, constructive criticism of our Introduction here.*

*To address (a), we rewrite original lines 21-24 to explicitly state how the cited works have contributed to including Indigenous knowledge in studying climate science or geologic hazards, not necessarily music (e.g., sea-level rise, earthquakes, volcanoes). To situate our study in traditional music, we then cite the success of works that have used Indigenous visual and performing arts to scientifically analyze geologic and environmental phenomena.*

> *Line 20 - 27: "Indigenous knowledge of plant, animal, and weather cycles have since been examined for signals of climate change and fortifications against climate hazards (e.g., Harmon et al., 2021; Hinzman et al., 2005; Hiwasaki et al., 2015; Turner and Reid, 2022). Workers also utilized Indigenous oral histories (e.g., Cascadia earthquakes in Ludwin et al., 2005), written records (e.g., Hawaiian hurricanes in Businger et al., 2018), and other sources of gray knowledge (e.g., various hazards of Aotearoa New Zealand in Bailey-Winiata et al., 2024) to improve modern-day disaster preparedness. Notably, scientists and cultural practitioners have focused on Indigenous visual and performing arts to examine historical volcanic activity (Swanson, 2008), earthquakes (Hough, 2007; Ludwin et al., 2005), and ecological resources (Gibson and Puniwai, 2006; Turner and Reid, 2022)."*

*Furthermore, we cite new literature that exemplifies how incorporating multiple ways of knowing (i.e., that recorded in Indigenous art) is beneficial for science by increasing research objectivity and creativity (Bang et al., 2018; Am. Acad. Arts Sci.; Intemann, 2009; Social Epistemology). Then, we end with a list of how further work in this field can benefit geoscience to prime our readers for the importance of our specific work examining Ryukyuan music.*

> *Line 28 - 32: "Expanding the scientific analyses of artistic traditions to more Indigenous cultures can increase research objectivity and creativity by introducing new questions from these marginalized viewpoints (Bang et al., 2018; Intemann, 2009). Thus, continued efforts in geoscience to incorporate Indigenous knowledge from art have great potential to 1) document more historical records of climate and geological phenomena, 2) jumpstart new collaborations between cultural practitioners and geoscientists, and 3) diversify scientific ways of knowing through the integration of Indigenous traditions."*

As for (b), four articles are cited as highlighting "previous works highlight how incorporating indigenous stewardship in Earth science courses can increase engagement by emphasizing the local, land-human interactions to which a learner has an ancestral connection" but we have no idea what these previous works have said on this topic.  Thus, it is unclear to the reader how this paper contributes to the literature on (a) and (b).

*We acknowledge the lack of background from the cited papers in this section. To accommodate point (b), we specifically introduce the sources of indigenous knowledge used by Chinn et al.*

*(2014) and Gibson and Puniwai (2006) to increase geoscience engagement in Hawai'i, similar to works done in New Mexico, USA, by Reano and Hasara (2024) and Reano and Ridgway (2015). We also describe the positive outcomes of including these multiple ways of knowing described in Alexiades et al. (2021). We then add Palmer et al. (2009; Journal of Geoscience Education), that utilized North American Indigenous art in an undergraduate course to showcase how art can increase geoscience engagement, which is relevant to our work here.*

> *Line 35 - 43: "For example, workers in Hawai'i have tested how geoscience classes can increase engagement with students by incorporating local indigenous stewardship, elder knowledge, Hawaiian place name meanings, and Hawaiian language newspaper records into their curriculum (e.g., Chinn et al., 2014; Gibson and Puniwai, 2006). Efforts in the Acoma Pueblo community of New Mexico, USA, have similarly integrated place-based concepts into education to teach about local stratigraphy, hydrology, and natural resources (e.g., Reano and Hasara, 2024; Reano and Ridgway, 2015). Furthermore, Palmer et al. (2009) focused on Indigenous art from the Southern Great Plains of North America to teach undergraduate science modules across mineralogy, groundwater, and climate hazards. These educators have demonstrated how rigorous programs that incorporate multiple ways of knowing can increase the interest, participation, and retention of students from marginalized communities (Alexiades et al., 2021)."*

Similarly, it is unclear from the brief mentions of the literature at the end of p.2 to the first paragraph on p.3, what has been written so far (if any) on the topic of Ryukyuan classical music and climatic/geological processes.  Is this paper filling a gap in any literature?  If so, what exactly are the gaps?

*Thank you for pointing out this deficit in our manuscript. We now discuss the background behind current work that examines Ryukyuan indigenous knowledge on scarce island groundwater (Takahashi, 2022), regional fisheries (Toguchi et al., 2016), and coral reef geobiology (Toguchi and Nishime, 2013).*

> *Line 46 - 48: "Previous works have connected Ryukyuan Indigenous knowledge with broad geoscience topics such as groundwater resources (Takahashi, 2022), regional fisheries (Toguchi et al., 2016), and coral reef geobiology (Toguchi and Nishime, 2013)."*

*We also explicitly state that there is a gap in researching Ryukyuan classical music in terms of geoscience engagement in both English and Japanese literature (to our best knowledge) despite Ryukyuan classical music being a very visible component of Okinawan culture today.*

> *Line 48 - 51: "However, to the best of our knowledge, no works in English or Japanese have examined the connections between geoscience and one of the most influential Indigenous art forms across Okinawa and the Okinawan diaspora, Ryukyuan classical music, also known as Ryūkyū koten ongaku (琉球古典音楽; hereafter RKO)."*

The methods is another section where scientific rigour could be improved; the methods are not described comprehensively.  In particular, it would be good for the reader to know (1) what are the other versions/schools of Nubui Kuduchi and Kudai Kuduchi exist, why the version used for

analysis in the paper was used (is it the "mainstream" version?  Why was it not compared with other versions/schools?);

*Thank you for this constructive comment towards improving our Methods section.*

*To address point (1), we now add sentences on why we opted to use the version of the songs here. As members of the mainstream Afuso Ryu school of Ryukyuan classical music, our lineage only covers the versions that we present here. However, J.Y. Uyeunten and K.A. Odo are also knowledgeable about the one other mainstream school called Nomura Ryu. The differences between schools are mainly in how the song is executed through singing and playing; there are only a few differences in lyrics, and the most relevant difference regarding volcanoes is already noted in the Results section. This overall similarity is likely because both schools diverged from the same lineage relatively recently in the 1800s. Gillan (2012) and new citation Garfias (1993; Asian Music) discuss how Afuso Ryu preserved the style of the original lineage while Nomura Ryu simplified the style to make the art more accessible. Therefore, we now explicitly discuss these different versions, the differences between versions, and why we use our version without comparison to other versions.*

> *Line 183 - 191: "In addition, we acknowledge that the other major RKO school, Nomura Ryū (野村流), may hold different versions of Nubui Kuduchi and Kudai Kuduchi. However, as all authors are members of an Afuso Ryū branch, we opt to use our school's version as a base, which is supplemented by insights from other textual and academic sources (i.e., Kinjo, 1992; Sakiyama and Oshiro, 1995; Seki, 2024). We note that Afuso Ryū and Nomura Ryū diverged from two students of the same Master Instructor in the 19th century and that most differences between the schools occur in singing or playing style; Afuso Ryū techniques are said to be closer to the original lineage, whereas those of Nomura Ryū were simplified and standardized to make the art more accessible (Garfias, 1993; Gillan, 2012). As the lyrics are mostly consistent across schools (except where we indicate notable variations), differences in interpretation due to version control are likely minimal."*

(2) a description of the Okinawan diaspora in general, and the diaspora in Hawai'i, …

*To address point (2), instead of in the Methods, we expand our newly added Introduction Section 1.1 to describe the Okinawan community in Okinawa Prefecture, in the diaspora, and the differences between both groups. We add a review on where diasporas are located, the number of immigrants, what kind of labor attracted them to go overseas, discrimination faced overseas, and the distinct Okinawan identity retained through 3+ generations, with a focus on the one in Hawai'i (e.g., population, affinity groups, cultural events).*

> *Line 87 - 108: "Despite this marginalization, the Ryukyu sphere of influence expanded out of East Asia during 19th–20th-century emigration from Okinawa Prefecture, namely*

*to Hawai'i, Brazil, and Peru that received roughly 20,000, 15,000, and 11,000 immigrants by 1938, respectively (Sellek, 2003). Estimates place cumulative Okinawan immigration between 150,000–200,000 people before the end of World War II; immigrants to the Japanese mainland were mostly comprised of factory workers, whereas those abroad worked on plantations (Roberson, 2010; Sellek, 2003). In Japan, Okinawans faced a similar system of discrimination as in Okinawa, compounded by social isolation, low wages, and dangerous working conditions (Roberson, 2010). Migrants overseas faced discrimination from two sources: poor plantation conditions by the elite-class plantation owners and a previously established, Japanese immigrant community that held similar prejudices as in Japan (Kaneshiro, 2002; Kodama, 1981; Ueunten, 1989). In Hawai'i, such racial tensions continued until roughly the end of World War II, when second-generation Okinawan and Japanese Americans became dominant over the first-generation immigrants; shared experiences in plantation labor unions, American military service, and communal education likely led to the gradual relinquishment of former prejudices (Ueunten, 1989). Nonetheless, the initial separation of Okinawan and Japanese led to a distinct, Okinawan, and diasporic identity. This identity is evident in the establishment and success of the Hawai'i United Okinawa Association (HUOA), an amalgamation of ~50 affinity groups that support Okinawan culture and community in Hawai'i (Kaneshiro, 2002; Kodama, 1981; Ueunten, 1989). An example of HUOA's success is the annual Okinawan Festival, one of the largest cultural events in Hawai'i that attracts ~50,000 attendees in the 21st century (Taira, 2023). Brazilian- (Mori, 2003) and Argentine-Okinawans (Alonso Ishihara, 2022), as well as Okinawan communities across the USA (Okamura, 2022), founded similar associations. Moreover, HUOA and these other associations participate in the Worldwide Uchinanchu Festival hosted by Okinawa Prefecture (Uchinanchu means "Okinawan People" in one Ryukyuan language); overseas associations sent ~8,000 attendees in 2016 (Okamura, 2022). Such efforts are fueled by a "Born Again Uchinanchu" movement that encapsulates third and later generations of the Okinawan diaspora looking to reconnect with their culture and heritage (Chinen, 2025). Accordingly, Okinawan identity remains visible and active worldwide."*

… and what kind of differences would the reader need to be aware of, in terms of the Okinawans in Okinawa vs. the diaspora elsewhere (also related to (1) above, are there differences in Nubui Kuduchi and Kudai Kuduchi sung by Okinawans in Okinawa and those transmitted within the diasporas?);

*In the new Section 1.2, we add a review on the differences between Okinawans in Okinawa and abroad: mainly the language and culture barrier after separation from Okinawa by 3+ generations.*

*Line 146 - 149: "Second, RKO and other folk music genres serve as pillars of Okinawan identity for the Okinawan diaspora, separated from Okinawa by three or more generations in the 21st century and interested in ways to express their identity (Kaneshiro, 2002; Ueunten, 1989). These later generations may not understand Ryukyuan languages and RKO lyrics, which pose a barrier to accessing Ryukyuan Indigenous knowledge (Chinen, 2025)."*

*We also address how Ryukyuan classical music instructors in the diaspora are often trained in Okinawa or trained by instructors from Okinawa. This training ensures that the versions transmitted in the diaspora are the same as those sung in Okinawa Prefecture.*

*Line 149 - 151: "However, most RKO instructors in the diaspora have trained in or are from Okinawa Prefecture, increasing direct access to such knowledge from the homeland (Chinen, 2025; Kaneshiro, 2002; Miyashiro, 2018; Teruya, 2014; Ueunten, 1989)."*

(3) more details on methodology used to "link" observations of climatic, geological, and environmental processes, as the authors refer to Swanson's work. This may help the reader understand better how photos and images (Figures 2, 4) were used in the analysis.

*To address point (3), we expand on and organize our Methods Section 2 into two subsections: 1) the Indigenous sources of lyrics and how we created our lyrical synopses and 2) the specifics of how we performed our literature review to find the "links" between song and science.*

*In new Section 2.1, we move our background of Nubui and Kudai Kuduchi, the various sources of Ryukyuan indigenous knowledge, and modern literature used to supplement our synopses. This is where we discuss the various permissions we obtained from Ryukyuan arts practitioners, how we collaborate between art and science practitioners to ensure rightful recognition of Ryukyuan sources, and why we base our interpretation off one version of both songs (i.e., the lyrics we are interpreting are very similar across the two main Ryukyuan classical music schools, also quoted above; Line 183 - 191).*

*Line 157 - 191: "We focus on Nubui Kuduchi and Kudai Kuduchi, which were composed during the Satsuma Domain's rule over the Ryukyu Kingdom. These songs are usually attributed to the RKO master Yakabi Chōki (屋嘉比朝寄; 1716–1775 CE; surname first following Japanese naming convention; Gillan, 2012; Kinjo, 1992). Kuduchi (口説) refers to a subgenre of RKO with a distinctly Japanese, rather than Ryukyuan, seven-five beat structure (Kinjo, 1992) and often tells a chronological story (Seki, 2024). Then, nubui (上り) refers to "climbing up" to Satsuma and kudai (下り) to "climbing down" to Okinawa Island (Kinjo, 1992; Seki, 2024). Thus, these songs detail a Ryukyuan envoy's 18th-*

*century journey between the Ryukyu Kingdom and Satsuma Domain during the Japanese colonial period (Fig. 1). Such performances were historically reserved for entertaining Satsuma Domain officials in the Ryukyu Kingdom, where performers dance with a folding fan in each hand or a traveler's cane for Nubui Kuduchi or Kudai Kuduchi, respectively (Fig. 2a–b; Sakiyama and Oshiro, 1995). The dance represents different aspects of the envoy and has a relatively masculine connotation related to its brisk tempo, karate influence, and the harrowing journey itself (Kinjo, 1992). Both songs remain popular in 21st-century RKO performances for entertainment and cultural preservation purposes (Hanashiro, 2007).*

*Here, we create English synopses of Nubui Kuduchi and Kudai Kuduchi to scientifically interpret both songs. We utilize a version of these songs from the Afuso Ryū (安冨祖流) school of RKO (one of two major schools; Garfias, 1993; Gillan, 2012), alongside interpretations from Kinjo (1992), Sakiyama and Oshiro (1995), and Seki (2024). Following best practices in Younging (2018), the authors here include RKO Master Instructors June Y. Uyeunten and Kenton A. Odo (hereafter J.Y. Uyeunten and K.A. Odo, respectively) of the Ryukyu Koten Afuso Ryu Ongaku Kenkyu Choichi Kai USA (hereafter Choichi Kai USA), serving the Okinawan diaspora in Hawai'i, USA. Both authors provide access to oral and written information on Nubui Kuduchi and Kudai Kuduchi, including personal communications and interpretations from Clarence T. Nakasone (hereafter C.T. Nakasone; 1998) of the Hooge Ryu Hana Nuuzi no Kai Nakasone Dance Academy, also based in Hawai'i. In addition, the first author is an uta sanshin practitioner with Choichi Kai USA at the time of publication. We provide supplementary videos with song lyrics, translations, and interpretations from the above sources, with permission and production from J.Y. Uyeunten, K.A. Odo, and the aforementioned dance academy (Higa et al., 2024a, b). We caution that these supplements are solely to provide references for lyrical synopses; we do not claim intellectual property for the songs and lyrics and do not assert that these songs should enter the public domain or become gnaritas nullius ("no one's knowledge" in Latin; i.e., Younging, 2018). These precautions are to ensure that Indigenous knowledge is properly credited and utilized. In addition, we acknowledge that the other major RKO school, Nomura Ryū (野村流), may hold different versions of Nubui Kuduchi and Kudai Kuduchi. However, as all authors are members of an Afuso Ryū branch, we opt to use our school's version as a base, which is supplemented by insights from other textual and academic sources (i.e., Kinjo, 1992; Sakiyama and Oshiro, 1995; Seki, 2024). We note that Afuso Ryū and Nomura Ryū diverged from two students of the same Master Instructor in the 19th century and that most differences between the schools occur in singing or playing style; Afuso Ryū techniques are said to be closer to the*

*original lineage, whereas those of Nomura Ryū were simplified and standardized to make the art more accessible (Garfias, 1993; Gillan, 2012). As the lyrics are mostly consistent across schools (except where we indicate notable variations), differences in interpretation due to version control are likely minimal."*

*In new Section 2.2, we provide more details on our literature review to find the correspondence or "links" we found between song and science. Here, we describe the methodology of Swanson (2008), namely utilizing one source of a Hawaiian oral history/chant/dance, of which there may be many versions, to compare with contemporary scientific literature. We note that this situation is similar to ours, where there is more than one version of both songs, but the meanings are fundamentally the same. We summarize that Swanson (2008) finds correspondence between Western and Indigenous ways of knowing.*

> *Line 193 - 203: "We link observations of climate, geology, and the environment within Nubui Kuduchi and Kudai Kuduchi synopses to 20th- and 21st-century scientific studies by noting similarities and differences therein, similar to Swanson (2008). Swanson (2008) utilizes a Hawaiian oral history of the volcano deity Pele and her sister Hiʻiaka in combination with 1812 CE written records from European Christian missionaries to improve scientific interpretations of caldera formation at Kīlauea Volcano, Hawaiʻi. The version of the oral history examined in Swanson (2008) is believed to be the most unaffected by Western influences, but other versions are likely to be fundamentally similar, much like Nubui Kuduchi and Kudai Kuduchi are similar across RKO schools. Swanson (2008) presents a literature review on Kīlauea geochronology and stratigraphy to compare with the oral and written histories. It was found that radiometric dating, Hawaiian oral tradition, and written records agree that the extant Kīlauea Caldera likely formed between 1470–1500 CE; the caldera was previously thought to have formed in 1790 CE during an explosive eruption. A major discussion point of Swanson (2008) is that such a conclusion may have been accepted by the scientific community earlier if Indigenous knowledge had been seriously considered."*

What is written in sections 4.2-4.3 show that the lyrics and historical records concur; how was this information analysed to lead to the discussion in section 5.1?

*In the rest of Section 2.2, we describe how we applied the framework of Swanson (2008) and focused on a literature review that specifically aims to find 1) similarities or differences between observations in Nubui/Kudai Kuduchi and modern literature (e.g., concurring monsoonal winds and volcanic activity) and 2) how the climate and volcanism described in song and science can affect humans today (e.g., impacts of climate change on typhoons, air pollution from volcanic activity). Discussion Section 4.1 (previously Section 5.1) thus relates to addressing point 1) and how the concurrence between Ryukyuan observations in song and modern instrumented science*

*can lead to interesting and novel scientific inferences. Section 4.2 (previously Section 5.2) will relate to addressing point 2) by using these scientific links to connect environmental science issues in Okinawa with students through place-based learning.*

> *Line 204 - 216: "Here, we follow the methods of Swanson (2008) because we have a parallel aim to examine the correspondence between an Indigenous record and scientific literature. For our literature review, we cover 20th – 21st-century research across Ryukyuan climate and volcanology, two topics that we identify as likely recorded in Nubui Kuduchi and Kudai Kuduchi. We focus our review on the natural conditions of climate and volcanic systems that Ryukyuan envoys may have experienced during 18th-century travels. We also review potential impacts on 21st-century Okinawans by anthropogenic climate change or volcanic hazards. Next, we systematically extract lyrical observations from our synopses as either climate- or volcanology-related. We then subgroup these observations into specific phenomena or locations from the corresponding scientific literature to showcase similarities or differences between sources. Finally, we discuss novel scientific and cultural implications from such climate and volcanic links, which can be tailored to geoscience engagement in Okinawa Prefecture and abroad. We acknowledge that, unlike Swanson (2008), Nubui Kuduchi and Kudai Kuduchi do not point to a single geologic event, but rather to general climate or geologic conditions. Nonetheless, these generalized links can be used in lessons to relate RKO with environmental research. We therefore demonstrate the utility of Indigenous knowledge from RKO to increase geoscience engagement in Okinawan communities."*

*Following these changes, we reorganize the Results Section 3, where 1) the literature review constitutes a separate "result" followed by 2) the correspondence in climate or volcanic processes that we propose are found in song and science. Thus, the literature review work does not precede the methods that describe how and why the literature review was done.*

*Because this literature review is mostly the same as the first draft (although some text is rearranged), please see Sections 3.2.1 (Line 268 - 293) and 3.3.1 (Line 327 - 353) in the manuscript PDF for the full text moved to this new location.*

*Moreover, we add another figure showing a diagram of how specific verses are related to literature in our review. We cite reordered Figures 1, 3 (originally Figure 4), and 4 (originally Figure 2) to emphasize how the observable phenomena or visitable sites in those figures are related to each link.*

**(a) Climate**

**Voyages**
- Voyages from Okinawa to Kyushu during summer south-southwesterlies
  • Nubui Kuduchi verse 5

- Voyages from Kyushu to Okinawa in the ninth or tenth lunar month
  • Kudai Kuduchi verse 1

- Winter north-northeasterlies
  • Kudai Kuduchi verse 6

**Prayers**
- Prayers for safe travels (Fig. 3a–b), vows to return to specific landmarks (Fig. 3c–d)
  • Nubui Kuduchi verses 1, 2, 4, 6
  • Kudai Kuduchi verses 2, 8, 9

- Rough seas between Iheya and the Amami Islands
  • Nubui Kuduchi verse 7
  • Kudai Kuduchi verse 8

- Famously dangerous Tokara Strait
  • Nubui Kuduchi verse 7
  • Kudai Kuduchi verse 6

**East Asian Monsoon**

Voyages from Okinawa to Kyushu occurred around summer (Okinawa Prefectural Cultural Promotion Association, 2001; Toby, 1984) when the Boreal monsoon is characterized by south-southwesterlies (Dobby, 1945; Flohn, 1957; Fu et al., 1983; Ueda et al., 1995).

Voyages from Kyushu to Okinawa occurred between autumn to spring when the Boreal monsoon is characterized by north-northeasterlies (Dobby, 1945; Fu et al., 1983).

**Ocean conditions**

Western Pacific Ocean is a typhoon hotspot during summer to autumn months when envoys were active (e.g., Ito et al., 2020).

Envoy path has statistically high wave heights, partially due to typhoon activity (Wu et al., 2014; Young et al., 2011).

Kuroshio current interactions with atmosphere and bathymetry in the Tokara Strait may induce ocean turbulence (Fig. 1; Hwang, 2005; Nagai et al., 2021; Tsutsumi et al., 2017).

[Figure]

Shuri Kannondo (Fig. 3a)

[Figure]

Sogenji (Fig. 3b)

[Figure]

Miegusuku (Fig. 3c)

[Figure]

Cape Zampa (Fig. 3d)

**(b) Volcanism**

- Smoke is burning/rising from Satsuma Iwojima
  • Nubui Kuduchi verse 8

- Kaimondake comes into view and Sakurajima's majesty could be mistaken for Mount Fuji
  • Nubui Kuduchi verse 8

**Satsuma-Iwojima**

Young ages of Iwodake lava (Kawanabe and Saito, 2002) and observed 20th–21st century fumaroles and ash eruptions (Kamada, 1964; Kazahaya et al., 2002; Shinohara et al., 2002; Fig. 4b).

**Sakurajima and Kaimondake**

The unincised Kaimondake edifice (Fujino and Kobayashi, 1997) is more geomorphically similar to Mount Fuji than Sakurajima (Fig. 4c–e).

Lack of historical recorded eruptions during the time of the envoy precludes that this mismatch is due to a catastrophic Sakurajima eruption (Fig. 4d–e; Kobayashi and Tameike, 2002).

[Figure]

Sakurajima (Fig. 4e)

*Line 295 - 299: "Figure 5: Linking lyrical observations with scientific literature on climate and volcanic systems that Ryukyuan envoys may have experienced. First, we extract (a) climate and (b) volcanic observations from Nubui Kuduchi and Kudai Kuduchi (top). Then, we subgroup lyrics that correspond with scientific literature on the East Asian Monsoon, ocean conditions, Satsuma-Iwojima, or Sakurajima and Kaimondake, related to oceanic*

*currents (Fig. 1), landmarks (Fig. 3), and volcanoes (Fig. 4) observable or visitable in the 21st century (bottom)."*

Sections 5.2 and 5.3 appear to be literature review sections, much of which could be brought to the Introduction section to show us the gaps (or, what this paper builds on) in the literature. Once again, the authors would need to include more details so the reader can get a better understanding of the literature on the topic.

*Thank you for pointing out this needed reorganization. We have moved much of the original Discussion sections to the Introduction to highlight the gaps and demand for our work in Ryukyuan indigenous knowledge. We also expand upon our discussion of the Ryukyu Kingdom, Ryukyu colonization and discrimination, the Okinawan diaspora, Ryukyuan classical music, and the environmental themes likely recorded in Ryukyuan classical music that could be useful for geoscience engagement.*

*In Section 1, we move up the previous work in Ryukyuan Indigenous knowledge and geoscience (i.e., moved the second paragraph in the original Section 5.2 and citations Takahashi, 2022; Toguchi et al., 2016; Toguchi and Nishime, 2013, as suggested) and the gap in utilizing Ryukyuan classical music in geoscience, as discussed and quoted in a previous response (Line 46 - 48).*

*In the new Section 1.1, we provide more details on the Ryukyu Kingdom, its conquest and annexation by Japan, assimilation policies implemented by Japan, and post-World War II history that led to revitalization efforts of Ryukyuan culture (i.e., moved much of the original Section 5.3 on Empowering Okinawan Communities to the Introduction, as suggested).*

> *Line 69 - 86: "The Ryukyu Islands span a north-south transect between Kyushu and Taiwan in the western Pacific Ocean, encompassing the Osumi, Tokara, Amami, Okinawa, Miyako, and Yaeyama islands at its maximum geographical extent (Fig. 1). This island arc is the ancestral home of the Indigenous Ryukyuan People and the former Ryukyu Kingdom, established in the 15th century and centered on Okinawa Island (Sakiyama and Oshiro, 1995; Toby, 1984). The kingdom colonized south to the Yaeyama Islands and north to the Tokara Islands during the height of this dynastic period (Akamine, 2017). Contact from foreign trade influenced Ryukyuan culture, including from China, Japan, Korea, Thailand, Malaysia, and Indonesia (Sakiyama and Oshiro, 1995). However, in 1609 CE, the Ryukyu Kingdom was invaded and subsequently controlled by Japanese forces in the Satsuma Domain of southern Kyushu and the Tokugawa Shogunate in Edo (pre-1868 CE name for Tokyo; Akamine, 2017; Toby, 1984). Consequently, most Ryukyuan territory north of Okinawa Island was ceded to the Satsuma Domain (Akamine, 2017). Historical records suggest ~20 Ryukyuan envoys traveled between Okinawa, Satsuma, and Edo to pay tribute to the Shogunate from 1610 CE until 1872 CE (Okinawa Prefectural Cultural Promotion Association, 2001; Toby, 1984), followed by the annexation of the Ryukyu Kingdom by Japan as Okinawa Prefecture in 1879 CE (Akamine,*

*2017). During this colonial period, the Japanese government employed assimilationist education policies with the goal of eliminating Ryukyuan languages and cultures (Hammine, 2019; Kaneshiro, 2002). This policy lasted until 1945 with the end of World War II and the start of occupation by the USA (Hammine, 2019). Since the reversion of Okinawa Prefecture from the USA back to Japan in 1972, Ryukyuan culture experienced a resurgence across grassroots movements (e.g., Inoue, 2004), language revitalization efforts (e.g., Heinrich, 2018; Zlazli, 2021), and statements from the Okinawan prefectural government (e.g., Abe, 2023)."*

*Then, we include more details on the Okinawan diaspora as highly active and distinct from the Japanese diaspora. We highlight that this Okinawan diaspora is another audience for the work done here due to historical discrimination and a demand to reconnect with their Okinawan heritage. This portion is quoted above in a previous response (Line 87 - 108).*

*In the new Section 1.2, we also provide more details on Ryukyuan classical music. Specifically, we include how Ryukyuan classical music is still relevant in Okinawa today due to the support of classical music schools, instructors, and organizations, as well as prefectural government support, quoted below.*

*Line 114 - 135: "RKO is one of many visible cultural identifiers for Okinawans in Okinawa Prefecture and in the diaspora (Gillan, 2016; Kaneshiro, 2002; Teruya, 2014; Ueunten, 1989, 2020). According to Sakiyama and Oshiro (1995), RKO is an aristocratic genre that developed during the Ryukyu Kingdom's dynastic period for entertaining visiting emissaries, historically accompanied by male dancers from the noble class (Fig. 2a–b). RKO lyrics are originally from Ryukyuan poetry, which often focuses on metaphors of the natural world to convey human emotions and experiences. These performances are led by uta sanshin (唄三線), or a three-stringed lute with vocals (Fig. 2c). The sanshin lute itself was brought to Okinawa from China and became a symbol of Okinawan identity (Gillan, 2016). Nonetheless, the uta vocal component holds the melody of most RKO songs and is often said to be more central to RKO than sanshin (Ueunten, 2020). Other instruments that accompany uta sanshin include fwansō (笛; bamboo flute; Fig. 2d), kūchō (胡弓; fiddle; Fig. 2e), kutū (箏; zither; Fig. 2f), and tēku (太鼓; drums; Fig. 2g). Such RKO gained distinct Japanese influences after the Satsuma invasion and increased contact with the Satsuma Domain through Ryukyuan envoys (Okinawa Prefectural Cultural Promotion Association, 2001; Toby, 1984). RKO developed into commercial and popular theater when the demand for Ryukyuan court music collapsed after Japanese annexation (Gillan, 2016), which ended historical class and gender restrictions in these aristocratic arts. As such, 21st-century RKO performing arts schools have wide participation in Okinawa Prefecture and the Okinawan diaspora; these schools remain*

*the main mode of RKO transmission to new learners (e.g., Gillan, 2016; Hanashiro, 2007; Kaneshiro, 2002; Ueunten, 1989). RKO is also visible to non-artists as stone monuments to transformative songs, called kahi (歌碑). These monuments are often installed where songs have some lyrical or historical connection to a place, functioning as community centers, artistic venues, and memorials to collective Okinawan experiences (e.g., World War II) or as landmarks in popular kahi tours (Gillan, 2017). Furthermore, RKO gained national and prefectural support through designations of National Living Treasures by the Japanese government and the establishment of institutions such as the Okinawa Prefectural University of Arts and the National Theatre Okinawa (Gillan, 2016). Thus, RKO remains a vibrant marker of Ryukyuan culture across the Okinawan community."*

I also recommend that the authors state the argument of the article clearly in the introduction. Currently, the first time the argument appears (after the abstract) is at the end of the methods section. It would be helpful for the reader to see this earlier.

*Thank you for pointing this out. We will more clearly state the argument of our paper in the Introduction.*

*Line 60 - 62: "Accordingly, we showcase how popular Indigenous music can document scientific observations of climate and geology to engage Indigenous Peoples with their contemporary environmental heritage."*

*Later, we are more specific in how this work will address Okinawan needs as:*

*Line 152 - 154: "As such, our investigation of RKO can fulfill geoscience engagement goals in Okinawa Prefecture and the global diaspora by elevating place-based, Ryukyuan, Indigenous knowledge as a reputable way of knowing."*

Once again, the fact that the authors have experiences and expertise as Ryukyu classical musicians makes this paper particularly appealing. Once the scientific rigour is strengthened, the paper could make an important contribution to scientific progress within the scope of Geoscience Communication.

*Thank you again for your helpful and constructive feedback for improving the scientific rigor of our manuscript.*

*Dear Reviewer 2,*

*Thank you very much for your comments on our work and support for artistic sources in science.*

The work is very interesting and includes meteorological and geological descriptions found in classical music pieces: Nubui Kuduchi and Kudai Kuduchi, two Ryūkyū koten songs composed during the rule of the Satsuma Domain over the Ryukyu Kingdom.

*We greatly appreciate your positive assessment and interest in our work on Ryūkyū koten.*

These songs provide valuable information for extracting historical climate data from Ryūkyū koten lyrics, which can be used to connect cultural arts with science and education in the western Pacific region.

The voyages described to the north of Okinawa Island, where numerous active volcanoes are located, offered a rare opportunity to document volcanism in Ryukyuan songs.

The use of artistic sources to obtain scientific information is an extremely rich field that deserves further exploration across many artistic disciplines.

The article adopts a multidisciplinary approach, highlighting aspects of historical, cultural, environmental, climatic, and geological interest.

I believe this is the right approach to enhance sources that deserve attention, also for educational purposes.

*Thank you for this very encouraging comment and accurate summary of our work.*

The introduction and the research objectives are clear; however, the geographical distinction between the Kyushu Islands and Okinawa is not clear, either in the text or in Figure 1.

*We appreciate this suggestion and can add a boundary between the Ryukyu island arc groups and Kyushu in Figure 1. We can draw a dashed line off the southern coast of Kyushu and label Kyushu as "Kyushu Island," similar to the other island groups for consistency. We can also edit the first sentence of the caption as "Figure 1: The Ryukyu Islands and southern Kyushu…" for geographical accuracy.*

[Figure]

*Line 65 - 68: "Figure 1: The Ryukyu Islands and southern Kyushu. Map shows the Kuroshio Current (Gallagher et al., 2015), Ryukyu Trench (Kamata and Kodama, 1994), approximate route of Ryukyuan envoys (Okinawa Prefectural Cultural Promotion Association, 2001), and subaerial Quaternary volcanoes (Global Volcanism Program, 2024). (inset) Detailed map of Okinawa Island. Geography from U.S. Department of State, Office of the Geographer (2013)."*

*We hope this makes the first in-text description of the Ryukyu Islands' extent and first citation of "Fig. 1" clearer regarding the distinction of Kyushu versus other island groups.*

*Line 69 - 71: "The Ryukyu Islands span a north-south transect between Kyushu and Taiwan in the western Pacific Ocean, encompassing the Osumi, Tokara, Amami, Okinawa, Miyako, and Yaeyama islands at its maximum geographical extent (Fig. 1)."*

Figure 2: Please add the meaning of the red dashed line in the caption.

*Thank you for this comment. We will add a legend for red dashed shapes as "Caldera or topographic depression" on Figure 2a (currently reordered as Figure 4a), following the journal guidelines that suggest against in-caption descriptions of figure items.*

[Figure]

*Line 260 - 265: "Figure 4: (a) Calderas and topographic depressions of southern Kyushu and the northern Ryukyu island arc (Maeno and Taniguchi, 2007; Nagaoka, 1988) with location of subplots as boxes or arrows indicating view direction. Geography from U.S. Department of State, Office of the Geographer (2013). PlanetScope 3 m resolution images of (b) Satsuma-Iwojima, (c) Kaimondake, and (d) Sakurajima (Image © 2025 Planet Labs; Planet Team, 2025) with 100 m elevation contours from 1 arc-second Shuttle Radar Topography Mission. (e) Sakurajima with Kagoshima City in foreground looking east versus (f) looking north with PlanetScope image draped over elevation."*

Each figure should be placed immediately after it is first mentioned in the text.

*We will rearrange the figures closer to their first mention in our revised text, as described in the journal guidelines.*

*Thank you again for your positive review of our manuscript. We are grateful for your enthusiasm regarding our work.*